# Moroccan natural products for multitarget-based treatment of Alzheimer's disease: A computational study

**Fatima Zahra Guerguer[1], Amal Bouribab[1], El Mehdi Karim[1], Meriem Khedraoui[1], Fatiha Amegrissi[1], Yasir S. Raouf[2], Abdelouahid Samadi[2]\*, Samir Chtita[1]\***

**1** Laboratory of Analytical and Molecular Chemistry, Faculty of Sciences Ben M'Sik, Hassan II University of Casablanca, Casablanca, Morocco, **2** Department of Chemistry, College of Science, United Arab Emirates University, Al Ain, United Arab Emirates

\* samadi@uaeu.ac.ae (AS); samirchtita@gmail.com (SC)

**Data Availability Statement:** All relevant data are within the manuscript and its Supporting Information files.

## Abstract

Alzheimer's disease is a neurodegenerative disorder that impairs neurocognitive functions. Acetylcholinesterase, Butyrylcholinesterase, Monoamine Oxidase B, Beta-Secretase, and Glycogen Synthase Kinase Beta play central roles in its pathogenesis. Current medications primarily inhibit AChE but fail to halt or reverse disease progression due to the multifactorial nature of Alzheimer's. This underscores the necessity of developing multi-target ligands for effective treatment. This study investigates the potential of phytochemical compounds from Moroccan medicinal plants as multi-target agents against Alzheimer's disease, employing computational approaches. A virtual screening of 386 phytochemical compounds, followed by an assessment of pharmacokinetic properties and ADMET profiles, led to the identification of two promising compounds, naringenin (C23) and hesperetin (C24), derived from *Anabasis aretioides*. These compounds exhibit favourable pharmacokinetic profiles and strong binding affinities for the five key targets associated with the disease. Density functional theory, molecular dynamics simulations, and MM-GBSA calculations further confirmed their structural stability, with a slight preference for C24, exhibiting superior intermolecular interactions and overall stability. These findings provide a strong basis for further experimental research, including *in vitro* and *in vivo* studies, to substantiate their potential efficacy in Alzheimer's disease.

## Introduction

Alzheimer's disease (AD) is the most prevalent form of dementia among the elderly, marked by severe cognitive decline affecting memory, thinking, and behaviour. It represents a significant global public health challenge due to its rising prevalence and considerable socio-economic impact [1]. According to the World Health Organization (WHO), more than 55 million people are currently living with dementia worldwide, and this number is expected to rise to 139 million by 2050, primarily due to an aging population [2].

**Funding:** AS and YSR acknowledge the support of the UAEU through an internal Start-up grant 2023 (Grant Code G00004400) and an internal Start-up grant 2024 (Grant 12S156), respectively.

**Competing interests:** The authors have declared that no competing interests exist.

Several theories aim to elucidate the underlying biological mechanisms driving the progression of AD. The widely accepted amyloid cascade hypothesis posits that the accumulation of β-amyloid plaques in the brain plays a pivotal role in disease pathogenesis. These plaques arise from the abnormal cleavage of amyloid precursor protein (APP) by β-secretase (BACE-1) and γ-secretase enzymes, forming insoluble β-amyloid peptides that disrupt neuronal communication and trigger neuroinflammatory and neurotoxic responses [3]. Additionally, the presence of neurofibrillary tangles composed of hyperphosphorylated tau proteins, stabilized by glycogen synthase kinase 3 beta (GSK-3β), disrupts neuronal function by impairing the stabilization of essential microtubules crucial for intracellular transport [4]. Dysfunction of the cholinergic system is also implicated in AD, marked by significantly reduced levels of acetylcholine, a neurotransmitter critical for memory and learning, due to the loss of cholinergic neurons [5]. Acetylcholinesterase (AChE) and butyrylcholinesterase (BuChE) enzymes degrade acetylcholine, and their inhibition is an established therapeutic strategy to elevate acetylcholine levels in the brain [6]. Furthermore, oxidative stress, exacerbated by monoamine oxidase B (MAO-B), and chronic inflammation contribute to neurodegeneration by causing cellular damage through an imbalance between free radical production and the body's antioxidant capacity [7]. Chronic inflammation is triggered by the immune response to amyloid plaques and tau tangles, further exacerbating neurodegeneration [8]. Various medical approaches targeting these pathological processes have failed in preclinical or clinical trials, partly due to complex disease etiologies, an incomplete understanding of the disease's exact cause, and the multi-factorial nature of AD [9]. Currently available anti-Alzheimer's drugs predominantly include AChE inhibitors like donepezil, rivastigmine, and galantamine, alongside the NMDA receptor antagonist memantine, which targets glutamate excitotoxicity [10]. These medications address specific aspects of Alzheimer's pathology, providing symptomatic relief without addressing the underlying molecular drivers of disease comprehensively, which further underscores the urgent need for more nuanced and effective options [11]. Given that various factors contribute to cognitive decline in AD, adopting a multi-target directed ligand approach (MTDL) appears useful to address this complexity. Unlike traditional approach focused on a single molecular target, MTDLs aim to develop molecules capable of binding multiple receptors or enzymes implicated in the incidence and progression of this neurodegenerative disease [12].

Historically, drug discovery relied solely on the synthesis and analysis of chemical compounds, which was a time-consuming and expensive process. However, the emergence of molecular modelling through computational methods has fundamentally transformed this approach. Researchers can now employ sophisticated computer simulations to predict the effectiveness and potential pharmacological properties of a compound before investing resources into synthesis and complex *in vitro* assays. This computational approach not only expedites the discovery of novel chemical matter but also reveals deep insights into how particular functional groups impact the targeted biological activities. As a result, molecular modelling presents a more efficient and economical means to explore and refine potential drug candidates, thereby greatly bolstering the prospects for developing groundbreaking pharmaceutical therapies [13–15].

Natural product research has historically essential medications due to their diverse chemical profiles and unique biological activities. Medicinal plants have long been crucial sources of treatments, with their constituent bioactive compounds often inspiring a significant portion of modern drug design and development [16]. Recent studies emphasize this potential via compounds like resveratrol and berberine for the treatment of AD, given their ability to cross the blood-brain barrier and effectively target the central nervous system effectively [17]. Previous research also suggests that *Artemisia annua* extract improves cognitive deficits and counteracts Alzheimer's-related pathological changes [18].

The rich biodiversity of Morocco and its longstanding tradition of medicinal plant use, especially in the treatment of neurodegenerative diseases like Alzheimer's, offer significant therapeutic potential. *Artemisia herba-alba*, recognized for its antioxidant and anti-inflammatory properties, has emerged as a promising candidate for neuroprotective therapies [19]. These findings highlight the crucial need to systematically explore natural resources to develop new multi-target agents for combating Alzheimer's disease [20].

The present study establishes a vital connection between the urgent need for multi-target therapies for AD and the landmark advancements in computational modelling, and computer-aided drug design (CADD). Accordingly, a database of phytochemical compounds derived from endemic medicinal plants of Morocco, traditionally used for centuries, was compiled. These plants, renowned for their therapeutic properties against neurodegenerative diseases, cancer, and diabetes, were specifically selected for their neuroprotective, antioxidant, and anti-inflammatory effects-key characteristics for addressing the multifactorial complexity of AD. A computational study based on virtual screening via molecular docking was implemented, followed by additional tools such as ADMET (Absorption, Distribution, Metabolism, Excretion, and Toxicity) prediction, density functional theory (DFT), and molecular dynamics (MD) simulations. This research provided a comprehensive analysis of the pharmacological profiles of the studied compounds, including assessments of their potential efficacy, pharmacokinetic profiles, toxicity, and metabolic stability within biological systems. These integrated in silico strategies enable a thorough evaluation of multi-target ligands targeting key enzymes such as AChE, BuChE, MAO-B, BACE-1, and GSK-3β, thereby opening new avenues for the development of innovative treatments against AD.

## Material and methods

### Data sources

The studied database was compiled from recent publications, focusing on phytoconstituents isolated from various endemic plants across different regions of Morocco, known in traditional medicine for their therapeutic properties [21–45]. The 2D structures of the 386 compounds, extracted from twenty-two plants, were generated using ChemOffice software. These structures were then optimized using the MMFF94 force field and the steepest descent algorithm via the Avogadro molecular software (version 2022), before being saved in PDB files. S1 Table (Supporting Information) provides the 2D structures of the studied compounds along with information such as the family name, plant, compounds, and region of origin.

### Molecular docking analysis

Molecular docking, aimed at predicting the optimal binding position of a ligand within a target enzyme, is crucial for understanding potential intermolecular interactions and the free energy of binding of the selected ligand, and the overall stability of the resulting complex [46]. The 386 phytochemical compounds were docked against the enzymes AChE, BuChE, MAO-B, BACE-1, and GSK-3β, using the Schrödinger Release 2020–3: Maestro software suite, following this structured protocol:

### Protein preparation

The crystallographic structures of the five proteins AChE, BuChE, MAO-B, BACE-1, and GSK-3β, along with their co-crystallized small molecule inhibitors (ligands), were obtained from the Protein Data Bank (www.rcsb.org) using the PDB identifiers 1EVE, 4BDS, 2V5Z, 4XXS, and 3F88, respectively. Before conducting the analysis, the proteins were prepared

using the Protein Preparation Wizard tool. Issues such as atom overlaps, alternate positions, missing atoms, and incorrect atom types were identified and corrected by adding hydrogen atoms, adjusting atom positions, minimizing the structures, and fixing missing side chains. Additionally, water molecules located beyond 0.5 Å were removed to prevent interference with protein-ligand interactions during docking. The proteins were then further processed in preparation for the study.

## Ligand preparation

The previously optimized ligands were further prepared using the LigPrep tool under the OPLS3e force field. The library underwent energy minimization. The software was configured to generate a maximum of 32 conformers for each ligand, with ionization states generated at a pH of 7 ± 2.

## Molecular docking

The receptor grids for the five proteins were generated using Maestro's Receptor Grid Generation tool, based on the co-crystallized ligands of the protein structures. The 386 phytochemical compounds underwent docking against AChE, BuChE, MAO-B, BACE-1 and GSK-3β using three different Glide docking modes: High-throughput virtual screening (HTVS), standard precision (SP), and extra precision (XP). These modes vary in precision, speed, and scoring function. HTVS mode facilitates rapid compound screening, minimizing intermediate conformations and reducing final torsion refinement and sampling [47]. Glide SP strikes a balance between speed and precision, typically requiring around 10 seconds per compound for exhaustive sampling [48]. XP mode, focused on eliminating false positives and penalizing molecules with low binding affinity, was specifically chosen for the final docking stage to provide more accurate binding affinity predictions. This enhanced precision ensured that only the most promising candidates, based on their interaction profiles, were selected for further analysis [49].

For each target protein, the docking protocol utilized the same approach with a specific standard ligand, utilizing the Glide score as the criterion to evaluate library compounds. Donepezil (DN) served as the positive control for AChE (PDB ID: 1EVE) due to well-established inhibitory activity against this enzyme [50]. Respective co-crystallized ligands were employed as molecular standards for BuChE (PDB ID: 4BDS), MAO-B (PDB ID: 2V5Z), BACE-1 (PDB ID: 4XXS), and GSK-3β (PDB ID: 3F88).

## Drug likeness and ADMET prediction

The evaluation of pharmacokinetic properties plays a crucial role in the therapeutic validation of our selected compounds. These characteristics encompass essential aspects such as absorption, distribution, metabolism, excretion, and toxicity (ADMET) [51]. Using canonical SMILES, we utilized the Swiss ADME web server (http://swissadme.ch/) for a thorough analysis of drug-likeness, whilst also assessing adherence to Lipinski's Rule of Five (Ro5) for oral bioavailability. Additionally, the pkCSM server (https://biosig.lab.uq.edu.au/pkcsm/) was employed for a detailed evaluation of ADMET profiles, providing key insights to predict and understand the potential *in vivo* pharmacology of our compound list [52]. This integrated approach enables a rigorous computational assessment, facilitating the identification of compounds that meet both therapeutic potential and bioavailability requirements, while strictly adhering to common pre-clinical safety standards [53].

## Molecular quantum analysis

Density Functional Theory, a quantum mechanical approach, examines the electronic density $\rho(r)$ in relation to the wave function $\Psi(r_1\sigma_1, r_2\sigma_2, \ldots r_n\sigma_n)$, where r and $\sigma$ denote the spatial and spin coordinates of all electrons in the atom or molecule [54]. Data on the electronic state of a molecule are crucial for assessing its reactivity by identifying sites for nucleophilic attack or addition, elucidating the nature of the chemical matter under study. Global reactivity descriptors [55], such as band gap energy, electronegativity, molecular hardness, molecular softness and HOMO/LUMO orbital energies, are commonly used in DFT to evaluate chemical reactivity. This approach aims to understand the stability of compounds without inducing unwanted chemical reactions that could yield harmful byproducts to the human body [56]. In our study, an extensive quantum mechanical optimization via DFT was performed on selected compounds, previously optimized using the MMFF94 force field. This optimization was conducted using Gaussian [57], employing the DFT B3LYP method [58] and the high-level 6-311G (d, p) quantum basis set [59]. This combination allows for a detailed description of the electronic environment, incorporating polarization effects and electron interactions, which are crucial for understanding the chemical behaviour of the analysed compounds. Electronic state spectra, HOMO and LUMO orbital energies (EHOMO, ELUMO) were generated, thereby facilitating the calculation of important molecular descriptors (Fig 1).

## Molecular dynamics simulations

Molecular dynamics (MD) simulation plays a pivotal role in our study, elucidating dynamic molecular interactions, more representative of the physiological reality as compared to more static computational methods [60]. To achieve this, we employed Desmond, a cornerstone of

**Ionization potential**

$$IP = -EHOMO$$

**Electron affinity**

$$EA = -ELUMO$$

**Energy gap**

$$Egap = ELUMO - EHOMO$$

**Hardness**

$$\eta = \frac{IP - EA}{2}$$

**Softness**

$$\sigma = \frac{1}{\eta}$$

**Electronegativity**

$$\chi = \frac{IP + EA}{2}$$

**Chemical potential**

$$\mu = -\chi$$

**Electrophilicity index**

$$\omega = \frac{\mu^2}{2\eta}$$

**Fig 1. Calculation formulas for the main molecular descriptors used in the DFT analysis.**

the Schrödinger Release 2020–3: Maestro software suite, to investigate how our selected compounds interact with the five targets studied. Each simulation spanned 100 nanoseconds, prepared using Schrödinger's Protein Preparation Wizard whilst maintaining a consistent pH of 7.4. This time frame is sufficient to observe complex molecular behaviours while avoiding the risk of missing critical dynamic events that could occur with shorter simulations. The System Builder facilitated the creation of a realistic simulation environment, utilizing the TIP3P water model for solvation in a orthorhombic simulation box (10 Å × 15 Å × 20 Å) [61]. To ensure electrical neutrality, Na$^+$/Cl$^-$ counter-ions were added at a concentration of 0.15 M. The robust OPLS3e force field was instrumental in configuring our systems [62]. Following initial equilibration, simulations were conducted under various thermodynamic ensembles (NVT and NPT), employing tailored thermostats and barostats to stabilize temperature and pressure. MD simulations were subsequently executed over 100 nanoseconds. Throughout these simulations, meticulous scrutiny of stability parameters such as RMSD, RMSF and protein-ligand contacts was conducted. These comprehensive investigations shed light on binding affinities and the efficacy of our selected compounds against the five biological targets studied.

## MM-GBSA calculations

To determine the free binding energies, we conducted Molecular mechanics with generalised Born and surface area solvation (MM-GBSA) calculations through Schrödinger's Maestro interface. The MM-GBSA panel within the prime module facilitated the estimation of free binding energies for our selected compounds, with the enzyme included in the workspace and compound selections guided by project table data [63]. The (VSGB) solvation model and OPLS3e force field were employed to enhance the accuracy and reliability of our assessments [64]. These integrated approaches provided a comprehensive understanding of the binding affinities and predicted efficacy profiles of our selected compounds across the five studied targets (AChE, BuChE, MAO-B, BACE-1, and GSK-3β).

## Results and discussion

### Molecular docking

Prior to proceeding with the key docking experiments, a crucial preliminary step involved conducting re-docking simulations for positive control ligands co-crystallized with various proteins. The tested ligands included donepezil (E20) with AChE (PDB ID: 1EVE), tacrine (THA) with BuChE (PDB ID: 4BDS), SAG with MAO B (PDB ID: 2V5Z), SI5 with BACE-1 (PDB ID: 4XXS), and 3HT with GSK-3β. The re-docking results revealed good accuracy, with RMSD values of 0.41 Å, 0.06 Å, 0.78 Å, 0.14 Å, and 0.43 Å, respectively (Fig 2). These findings affirm the reliability of the docking protocol employed in our study.

Following these initial control experiments, we conducted an in-depth analysis of the docking interactions of 386 phytochemical compounds with the target proteins in our study. A comparison with the co-crystallized ligands is presented in the Table 1. Negative binding energy scores indicate increased stability of the ligand-protein complexes, with lower scores denoting stronger and more stable interactions. Our study specifically focused on 17 compounds selected from the initial list of 386, which demonstrated binding energies lower than those of the standard ligands for at least two of the five targets (AChE, BuChE, MAO-B, BACE-1, and GSK-3β). These compounds emerged as potential anti-Alzheimer agents due to their ability to simultaneously target multiple disease-implicated proteins with favourable free energies. Subsequently, these 17 selected compounds were subjected to further ADMET analysis to also evaluate their pharmacokinetic tractability.

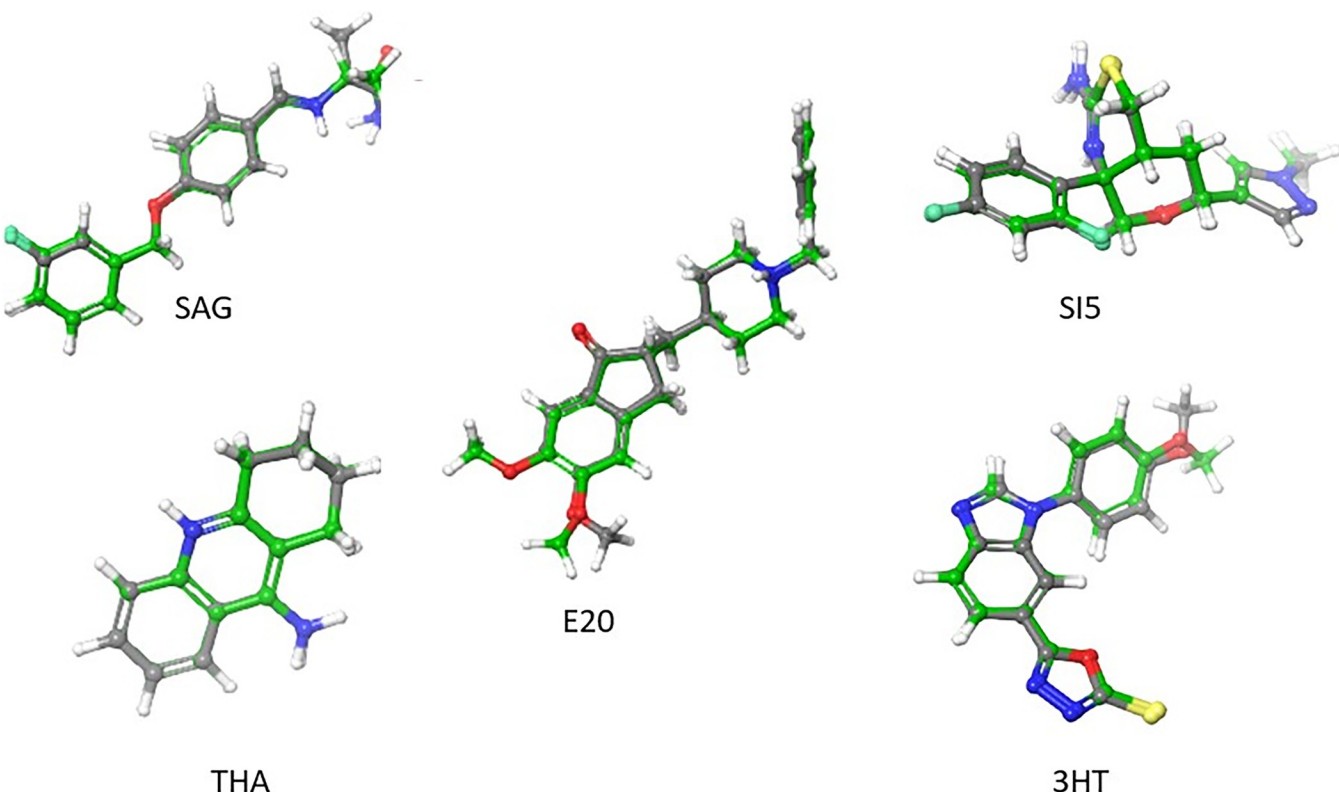

**Fig 2. Superimposed poses of the original (grey) and redocked (green) ligands within the protein receptor pockets.**

**Table 1. Binding affinity of the best complexes formed between docked compounds and target proteins (AChE, BuChE, MAO-B, BACE-1, and GSK-3β).**

| No | Name | Binding affinity kcal/mol | | | | |
|---|---|---|---|---|---|---|
| | | AChE | BuChE | MAO-B | BACE-1 | GSK-3β |
| | Co-crystallized ligand | *-12.57* | *-8.28* | *-8.78* | *-7.54* | *-7.48* |
| C2 | Catechin | -11.86 | -8.77 | -11.10 | -6.94 | -9.26 |
| C7 | Quercetin | -10.36 | -10.04 | -11.81 | -6.89 | -11.17 |
| C10 | epicatechin | -11.85 | -8.87 | -11.07 | -6.90 | -9.19 |
| C22 | Luteolin | -10.05 | -9.76 | -11.33 | -7.62 | -10.78 |
| C23 | naringenin | -9.92 | -8.42 | -10.48 | -6.53 | -9.11 |
| C24 | Hesperetin | -10.16 | -8.13 | -10.87 | -7.98 | -9.92 |
| C71 | 3-(2-N-Acetyl-N-methylaminoethyl) indol | -7.04 | -8.08 | -8.68 | -5.13 | -7.25 |
| C107 | p-Coumaroylquinic acid | -8.63 | -8.34 | -9.74 | -6.55 | -8.91 |
| C276 | Myricetin | -10.98 | -8.54 | -12.34 | -8.49 | -12.53 |
| C284 | Harpagid | -12.02 | -11.13 | -9.16 | -8.04 | -8.65 |
| C286 | Ferulic acid 4-O-glucoside | -11.44 | -7.29 | -10.05 | -8.44 | -6.69 |
| C288 | Epigallocatechin gallate | -13.86 | -11.29 | -11.92 | -7.92 | -8.98 |
| C289 | 8-O-acetyl-harpagid | -10.27 | -10.42 | -10.49 | -9.08 | -7.41 |
| C294 | Apigenin | -8.79 | -8.23 | -10.05 | -6.26 | -9.14 |
| C296 | Kaempferide | -9.01 | -8.48 | -10.90 | -6.18 | -9.13 |
| C321 | Luteolin-glucuronide | -12.17 | -8,24 | -12.02 | -8.05 | -5.97 |
| C329 | Jaceosidin | -11.16 | -6.84 | -11.79 | -7.24 | -9.59 |

## Drug likeness and safety profile prediction of selected compounds

Ensuring the safety and efficacy of drug molecules is paramount in drug discovery. In a recent survey of potential clinical challenges met during drug development highlighted inadequate clinical safety, dose-limiting toxicities, and idiosyncreatic (i.e., unpredictable) side-effects as key determinants for the termination of most drug discovery programs [65]. *In silico* pharmacokinetic profiling has proven to be a valuable method for evaluating these aspects in early development stages. In this study, we assessed the drug-likeness of small molecules based on Lipinski's Ro5, fundamental for initial pharmacological tractability and oral bioavailability. Various ADMET parameters (absorption, distribution, metabolism, excretion, and toxicity) were then analysed to determine the potential utility of these compounds for AD treatment. Among the initially selected 17 compounds, only compounds C23 and C24 were retained based on their *in silico* pharmacokinetic profiling.

The drug likeness assessment of compounds C23 and C24, as illustrated in Table 2, reveals promising characteristics suggesting their potential for drug development, compared to Donepezil, an FDA-approved drug for the treatment of AD. The molecular weights of C23 and C24 are 272.25 g/mol and 302.28 g/mol, respectively, both well below the 500 g/mol threshold set by Lipinski's Rule of Five. Similarly, Donepezil, with a molecular weight of 379.49 g/mol, also adheres to this limit. All three compounds demonstrate a favourable bioavailability score of 0.55. In terms of molecular flexibility, C23 and C24 exhibit low flexibility, with 1 and 2 rotatable bonds, respectively, while Donepezil possesses 6. This suggests greater rigidity for C23 and C24, which may be advantageous for stable interactions with target proteins. Regarding hydrogen bonding, C23 and C24 have 5 hydrogen bond acceptors and 3 donors, meeting the requirements of Lipinski's Rule, which stipulates fewer than 10 acceptors and fewer than 5 donors. In contrast, Donepezil has 4 acceptors and no donors, which may limit hydrogen bonding interactions but still aligns with the rule's criteria. With respect to lipophilicity, evaluated by the octanol/water partition coefficient (Log P), a Log P value below 5 is considered favourable for membrane permeability and solubility. C23 and C24 show Log P values of 2.5 and 2.51, respectively, indicating moderate hydrophilicity and compliance with the criteria. However, Donepezil has a Log P of 4.0, demonstrating greater lipophilicity, which may influence its absorption while remaining within acceptable limits. All compounds adhere to Lipinski's Rule of Five without any violations, and no alerts regarding pan-assay interference compounds (PAINS) have been detected, reinforcing their potential as drug candidates. Although C23 and C24 exhibit significant potential, it is essential to note that Donepezil is a clinically approved drug with a well-established safety and efficacy profile. Conversely, C23 and C24 still require experimental validation to confirm their therapeutic potential. Also,

**Table 2. Assessment of drug likeness for compounds C23, C24 and Reference drug based on Lipinski's criteria.**

| Compound | Drug likeness properties | | |
|---|---|---|---|
| | C23 | C24 | Reference drug Donepezil |
| MW (g/mol) | 272.25 | 302.28 | 379.49 |
| H-Bonds Acceptors | 5 | 5 | 4 |
| H-Bonds Donors | 3 | 3 | 0 |
| Log P | 2.50 | 2.51 | 4.00 |
| Rotatable Bonds (Flexibility) | 1 | 2 | 6 |
| Lipinski Violation | 0 | 0 | 0 |
| PAINS Alerts | 0 | 0 | 0 |
| Bioavailability Score | 0.55 | 0.55 | 0.55 |

**Table 3. ADMET profiles of compounds C23, C24 and reference drug.**

| ADMET | Properties | Compound | | |
|---|---|---|---|---|
| | | C23 | C24 | Reference drug Donepezil |
| Absorption | Water solubility (log mol/L) | -3.20 | -3.67 | -4.64 |
| | Caco2 permeability (log Papp in $10^{-6}$ cm/s) | 1.24 | 0.79 | 1.27 |
| | Intestinal absorption (human)% | 90 | 79 | 93 |
| Distribution | BBB Permeability (log BB) | -0.89 | -0.93 | 0.15 |
| | CNS Permeability (log PS) | -2.23 | -3.00 | -1.46 |
| Metabolism | CYP1A2 inhibitor | No | No | No |
| | CYP2C19 inhibitor | No | No | No |
| | CYP2C9 inhibitor | No | No | No |
| | CYP2D6 inhibitor | No | No | Yes |
| | CYP3A4 inhibitor | No | No | Yes |
| Excretion | Total clearance (log mL/min/kg) | 0.07 | 0.03 | 0.98 |
| Toxicity | hERG I inhibitor | No | No | No |
| | hERG II inhibitor | No | No | Yes |
| | Hepatotoxicity | No | No | Yes |
| | Skin sensitization | No | No | No |

while C23 and C24 did not present any structural alerts at this stage, flavonoids have been previously shown to act as PAINS compounds with false-positive potencies [66]. At the same time, these scaffolds are notoriously prone to form redox-cycling quinones motifs, with potentially potent yet unpredictable cellular reactivity. Therefore, both of these aspects must be considered in future experimental work, and if confirmed must be designed out via structural derivatization strategies.

The pharmacokinetic properties (absorption, distribution, metabolism, and excretion) and toxicity of the selected compounds have been rigorously evaluated to determine their potential as drug candidates, with the results summarized in the Table 3. Regarding absorption, high water solubility facilitates effective dissolution, while increased predicted permeability across Caco-2 cells indicates good intestinal absorption. Compounds C23 and C24 stand out with significant aqueous solubility values of -3.20 and -3.67, respectively, compared to Donepezil, which has a lower solubility of -4.64. The Caco-2 permeability values indicate that C23 (1.24) and C24 (0.79) possess reasonable permeability, while Donepezil shows slightly higher permeability at 1.27, suggesting more efficient intestinal absorption. However, the predicted intestinal absorption percentages (HIA %) for C23 (90%) and Donepezil (93%) are quite similar, whereas C24 exhibits moderate absorption at 79%. In terms of distribution, achieving effective penetration across the blood-brain barrier (BBB) poses a significant challenge, particularly for treating central nervous system (CNS) disorders. Compounds with a LogBB value below -1 demonstrate limited brain diffusion, while those exceeding 0.3 exhibit better permeability. Donepezil, with a LogBB value of 0.15, shows superior penetration across the BBB compared to C23 (-0.89) and C24 (-0.93), suggesting that the reference drug may offer slightly more effective brain diffusion. Nonetheless, the LogBB values of C23 and C24, being greater than -1, indicate their potential to cross the barrier, albeit to a lesser extent. Metabolically, the inhibition of cytochrome P450, which is crucial for hepatic drug metabolism, has been evaluated. The selected compounds do not inhibit the enzymes CYP1A2, CYP2C19, CYP2C9, CYP2D6, or CYP3A4, thereby reducing the risk of hepatic dysfunction and adverse drug interactions. In contrast, Donepezil inhibits both CYP2D6 and CYP3A4, potentially leading to undesirable metabolic interactions. Moreover, the predicted total clearance values are noteworthy, with

C23 and C24 displaying very low values (0.07 and 0.03, respectively) compared to Donepezil (0.98), suggesting that compounds C23 and C24 may have prolonged half-lives, although further studies are needed to confirm this potential advantage. Finally, toxicity assessments, including hepatotoxicity, cardiotoxicity, and skin sensitization, reveal a favourable safety profile for compounds C23 and C24. Neither compound exhibited signs of hepatic or cardiac toxicity or skin sensitization. In contrast, Donepezil poses a risk of hepatotoxicity and inhibition of hERG II channels, indicating potential cardiac and hepatic toxicity. This could confer a safety advantage to compounds C23 and C24 over the reference drug. While compounds C23 and C24 exhibit promising properties in terms of solubility, absorption, and a favourable safety profile compared to Donepezil, they show slightly lower brain penetration. However, their potential as drug candidates is further bolstered by their capacity to act as multi-target inhibitors. Consequently, more comprehensive experimental evaluations are necessary to gain a better understanding of their complete pharmacokinetic profiles and to assess their therapeutic potential in clinical applications.

## Interactions formed between selected compounds and targets

Based on the results of the molecular docking analysis and the prediction of the Drug Likeness and safety profile, two compounds were selected for their multi-target profile and favourable characteristics. Compound C23, known as naringenin and extracted from the plant *Anabasis aretioides*, was identified as a multi-target agent. It exhibited high affinity scores of -8.42, -10.48, and -9.114 kcal/mol compared to the reference compounds, which were -8.28, -8.783, and -7.483 kcal/mol for BuChE, MAO-B, and GSK-3β, respectively. Additionally, it showed notable affinity scores for AChE and BACE-1, with values of -9.922 and -6.537 kcal/mol compared to the reference compounds, which were -12.57 and -7.98 kcal/mol, respectively. Compound C24, known as hesperetin and also derived from the same plant, demonstrated a multi-target profile, exhibiting affinity scores of -10.871, -7.98, and -9.923 kcal/mol, surpassing the reference compounds, which had scores of -8.783, -7.54, and -7.483 kcal/mol for MAO-B, BACE-1, and GSK-3B, respectively. Moreover, hesperetin showed significant affinity scores for AChE and BuChE, with values of -10.161 and -8.132 kcal/mol, compared to the references of -12.57 and -8.28 kcal/mol. These results indicate that both compounds possess superior multi-target therapeutic potential, making them promising candidates for the development of treatments for AD. Consequently, these compounds were further evaluated based on their interactions with important amino acid residues, using the reference compounds for each protein as controls.

AChE features an active site composed of the residues Ser200, His440, and Glu327, forming a catalytic triad essential for acetylcholine hydrolysis. The catalytic gorge, including Trp84, Tyr121, and Phe330, stabilizes the enzyme-substrate complex through π-π and hydrophobic interactions [50]. The anionic subsite, involving Tyr70 and Asp72, facilitates the binding of the substrate's cationic head. The peripheral anionic site (PAS), comprising Trp279 and Tyr334, plays a crucial role in the recognition and modulation of inhibitors and substrates [67]. Docking analysis revealed π-cation interactions between the NH+ group of Donepezil and the residues Tyr334, Phe331, Phe330, and Trp84, along with a π-π stacking interaction with Trp279 and its aromatic ring. Additionally, a hydrogen bond between the ketone motif within Donepezil and the residue Phe288 was observed. Compounds C23 and C24, on the other hand, exhibited significant π-π interactions with the residues Phe331 and Phe330, respectively, via their aromatic rings. Furthermore, hydrogen bonds were also found with Arg269 through their free hydroxyl groups (Fig 3).

Regarding BuChE, the active site of this enzyme comprises a catalytic triad, a choline binding pocket, and an acyl binding pocket, all located within a 20 Å deep tunnel. The catalytic

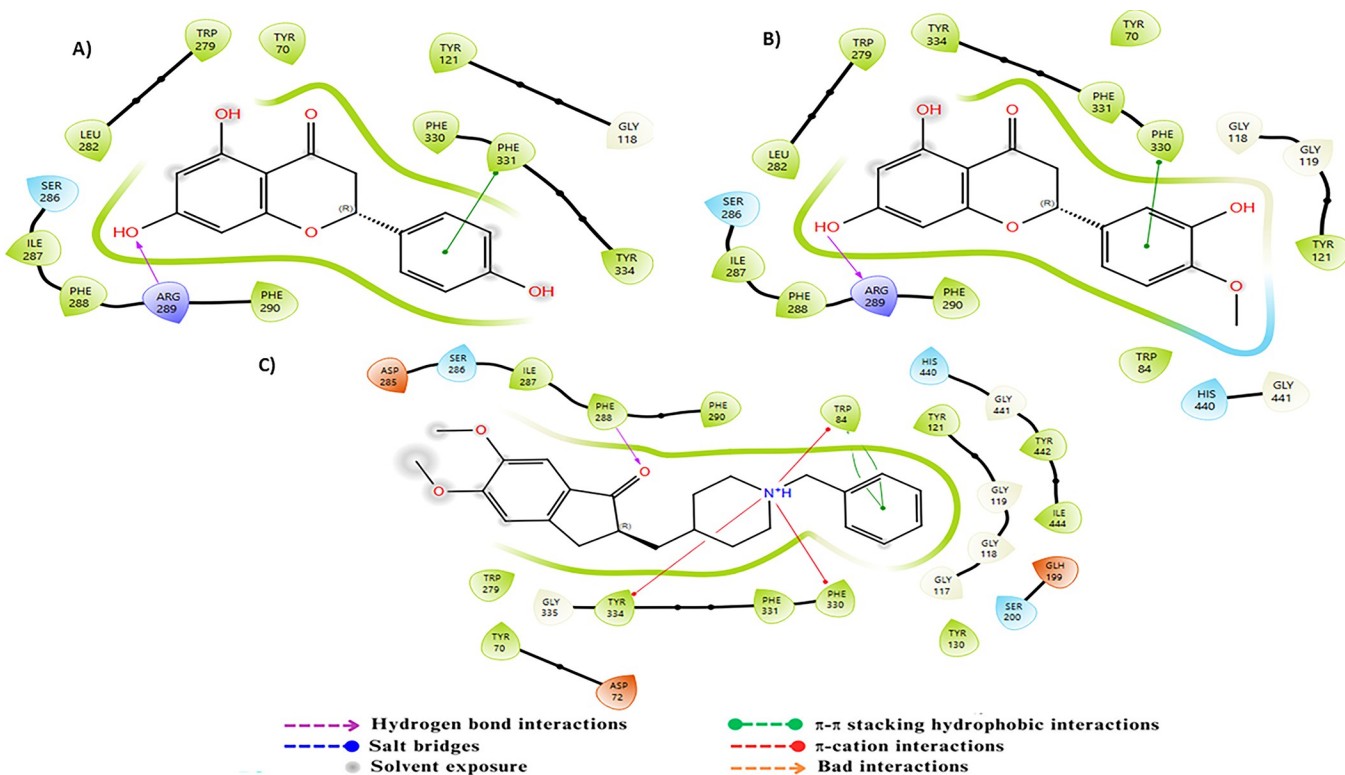

**Fig 3.** Two-dimensional (2D) representation of the docking poses in the active site of AChE (PDB ID:1EVE) for compounds A) C23, B) C24 and C) the reference.

triad consists of three key residues: Ser198, His438, and Glu325 [68]. Additionally, the residues Phe329 and Trp332 play a crucial role in attracting ligands into the gorge, and Asp70 and Trp82, located in the peripheral anionic pocket, also contribute to stabilizing ligand binding [69]. The primary interactions between the ligand tacrine and BuChE include π-aromatic stacking with Trp82, which facilitates the attraction of tacrine into the deep gorge, as well as a hydrogen bond between the amino group of tacrine and His438. Compound C23 demonstrated excellent interactions with BuChE by forming three hydrogen bonds with the residues Glu197, Tyr128, and Thr120, and engaging in π–π stacking with Trp82. Compound C24, on the other hand, showed remarkable interactions by forming three hydrogen bonds with Glu197, Asn83, and Hip438 via its different hydroxyl groups, as well as a π-π interaction with Trp82 via its aromatic ring. These complex interactions highlight the strong binding affinities of compounds C23 and C24 with BuChE (Fig 4).

For MAO-B, this enzyme is a flavoprotein composed of 520 amino acids forming two cavities: an entry cavity and a reactive cavity for substrate binding [70]. Research has demonstrated that amino acids Lys296, Trp388, Tyr398, and Tyr435, which form an aromatic sandwich at the substrate binding site, are essential for the catalytic activity of MAO-B [71]. Docking results of the standard ligand revealed that its amide amine group and NH+ group form two hydrogen bonds with residue Gln206. Furthermore, π-π stacking interactions were observed between the aromatic ring of the standard ligand and Tyr326. Compound C23 exhibits π-π stacking interactions with residues Tyr435 and Tyr395 via its aromatic ring, along with a hydrogen bond facilitated by its hydroxyl group. Compound C24 demonstrates π-π stacking between the aromatic ring of coumarin and residues Tyr398 and Tyr435, as well as another π-

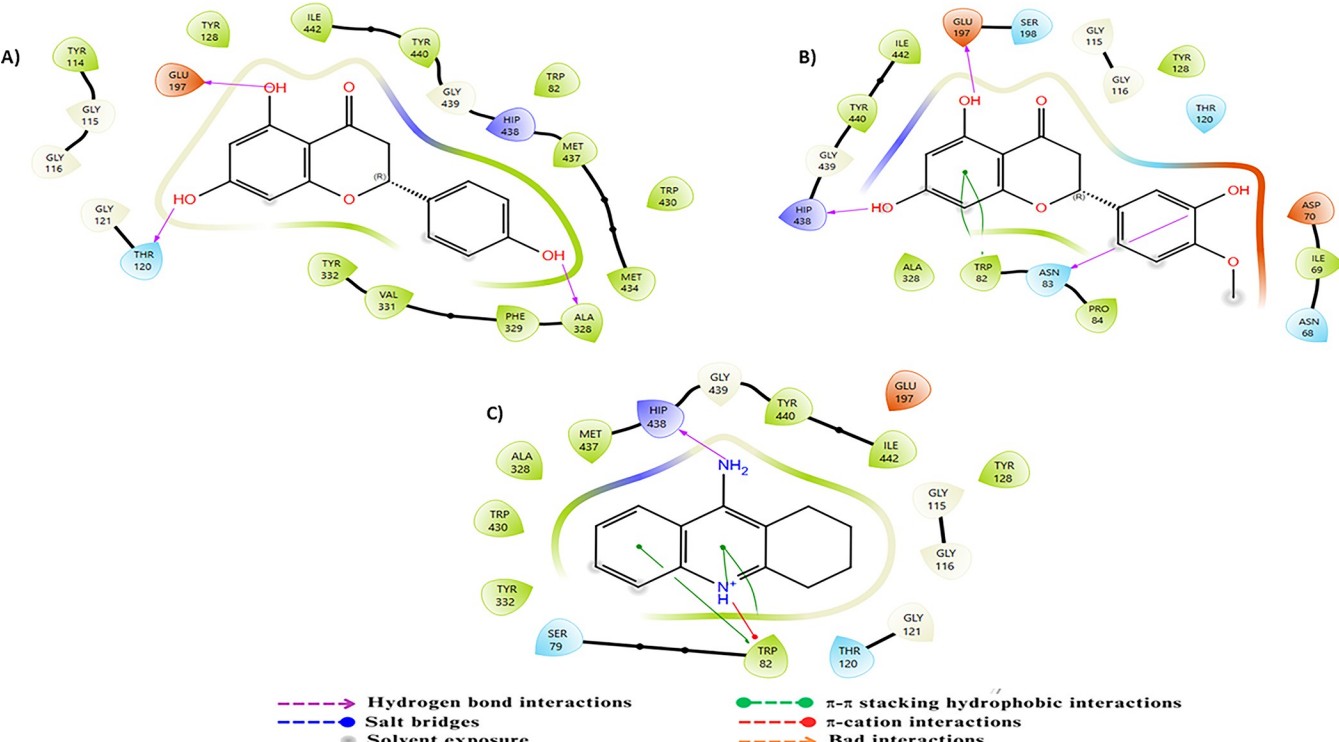

**Fig 4.** Two-dimensional (2D) representation of the docking poses in the active site of BuChE (PDB ID:4BDS) for compounds A) C23, B) C24 and C) the reference.

π stacking interaction with the aromatic ring. A hydrogen bond from the hydroxyl group to residue Gly434 is also observed. These intricate interactions underscore the strong binding affinities of compounds C23 and C24 with MAO-B, highlighting their significant inhibitory potential, particularly due to their interactions with essential residues involved in MAO-B inhibition (Fig 5).

For BACE-1, an aspartic protease, the active site consists of several distinct subsites: S1 containing the catalytic residues Asp32 and Asp228, S2 primarily composed of hydrophobic residues such as Ser35, Val69, Tyr71, Ile126, and Tyr198, and S3 and S4 which are solvent-exposed and include residues like Pro70, Thr72, Glu125, Arg128, Arg195, and Trp197 [72, 73]. The reference ligand establishes robust interactions with Asp32 via the N3 atom of the 1,3-thiadiazine ring, ensuring a solid anchoring in the binding site, along with a hydrogen bond between Asp32 and the amino group linked to the 1,3-thiadiazine ring. In contrast, compound C23 stands out due to significant interactions facilitated by hydrogen bonds between its hydroxyl groups and the residues Asn37 and Gly230, as well as between the ketone group and Trp76. Similarly, compound C24 shows notable interactions through hydrogen bonds between its hydroxyl groups and the residues Phe108 and Ser35, reinforced by π-π stacking with Trp76 through its aromatic ring. Additionally, the methoxy substituent of compound C24 effectively occupies the lipophilic groove formed by Arg128, Ala127, and Ile126, thereby enhancing its inhibitory potential. These complex interactions illustrate the ability of compounds C23 and C24 to specifically interact with key residues in the active site of BACE-1, suggesting their potential as promising inhibitors of this enzyme target (Fig 6).

GSK-3β, a member of the serine/threonine kinase family, utilizes ATP as its natural ligand and possesses three distinct binding sites: an ATP binding site consisting of Leu132, Tyr134,

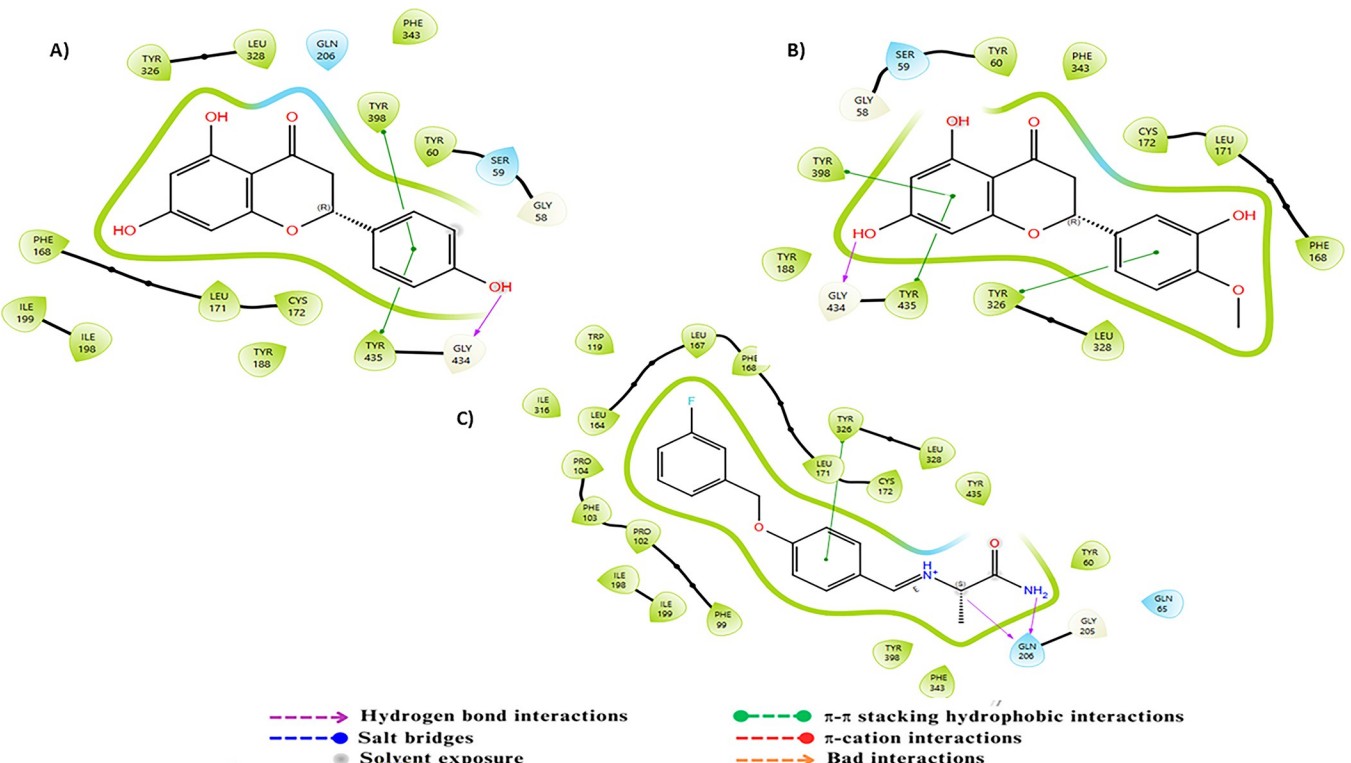

**Fig 5.** Two-dimensional (2D) representation of the docking poses in the active site of MAO-B (PDB ID:2V5Z) for compounds A) C23, B) C24 and C) the reference.

Val135, Pro136, and Arg141; an Axin binding site composed of Lys85, Asp133, Val135, Lys183, and Asp200; and a priming site including Arg96, Arg180, Ser203, Lys205, and Val214, [74, 75]. The reference ligand, a selective indazole-based inhibitor, forms a hydrogen bond between its NH+ group and Val135 in the hinge region of GSK-3β, as well as a hydrogen bond between its thione group and Lys85. Furthermore, the methoxyphenyl motif establishes a π-π stacking interaction with 2HT998. Docking results for compound C23 show that the coumarin ketone group forms a hydrogen bond with Val135, while its aromatic ring establishes a π-π stacking interaction with 2HT998. Similarly, compound C24 shows that its coumarin ketone group forms a hydrogen bond with Val135, and its aromatic ring engages in a π-π stacking interaction with 2HT998. Additionally, the hydroxyl groups attached to the coumarin nucleus and the methoxyphenyl form hydrogen bonds with Asp133 and Pro136, respectively. These two compounds, C23 and C24, exhibit key interactions with essential residues, surpassing even those of the reference ligand, thereby suggesting superior inhibitory potential (Fig 7).

## Molecular quantum analysis

Density Functional Theory (DFT) is an essential tool in studying the fundamental quantum states of molecular systems. It is particularly valuable for analyzing bioactive compounds, providing insights into their physicochemical properties, reaction sites, and potential effects [76]. In this study, DFT was employed to examine the electronic properties and chemical reactivity of the two selected compounds. The energy of HOMO reflects electron-donating ability of compound, indicating its stability. Compound C23 has a HOMO energy of -6.285 eV, while compound 24 has -5.988 eV. Conversely, the energy of LUMO indicates electron-accepting

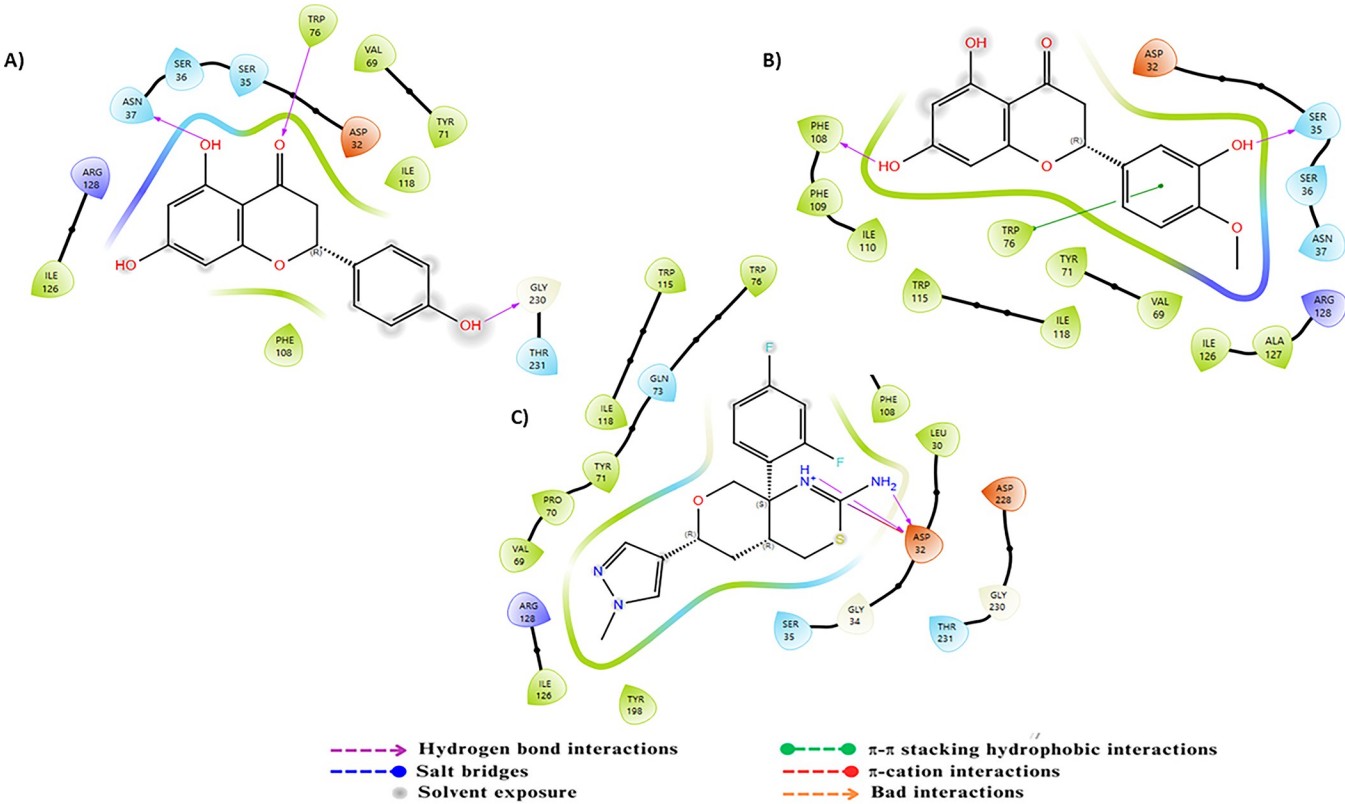

**Fig 6.** Two-dimensional (2D) representation of the docking poses in the active site of BACE-1 (PDB ID:4XXS) for compounds A) C23, B) C24 and C) the reference.

capacity, representing the excited state. Compound C23 has a LUMO energy of -1.660 eV, and compound C24 has -1.602 eV. The energy gap (Egap) between HOMO and LUMO energies is crucial for assessing a compound's propensity for electron transitions and chemical reactions. Compounds C23 and C24 exhibit energy gaps of 4.626 eV and 4.385 eV, respectively, suggesting low reactivity and high stability. This stability is further confirmed by negative values of chemical potential (μ) for both molecules (-3.972 eV and -3.795 eV), indicating spontaneous inclusion processes. Chemical hardness (η) is another important measure of chemical stability. Molecules with a large energy gap are considered "hard" and more stable. Compounds C23 and C24 show chemical hardness values of 2.313 eV and 2.193 eV, respectively, suggesting stability and resistance to chemical reactions.

Additionally, other electronic parameters were evaluated. Chemical softness (σ), the inverse of chemical hardness, describes a molecule's ease of polarization. Compound C23 has a chemical softness of 0.432 eV, while compound C24 has 0.456 eV. Electronegativity, which measures a compound's tendency to attract electrons, is 3.972 eV for compound C23 and 3.795 eV for compound C24, indicating slightly greater electron-attracting ability for compound C23. In terms of electrophilicity, compound C23 (3.411 eV) shows slightly higher electron-attracting potential than compound C24 (3.284 eV), which can be critical for interactions with electrophilic sites. The dipole moment, an indicator of compound polarity, is higher for compound C24 (4.395 Debye) compared to compound C23 (2.603 Debye), indicating higher polarity for C24. Lastly, the total electronic energy, lower for C24 (-1069.758 eV) compared to compound C23 (-955.203 eV), reveals increased energy stability for compound C24. Table 4 summarizes

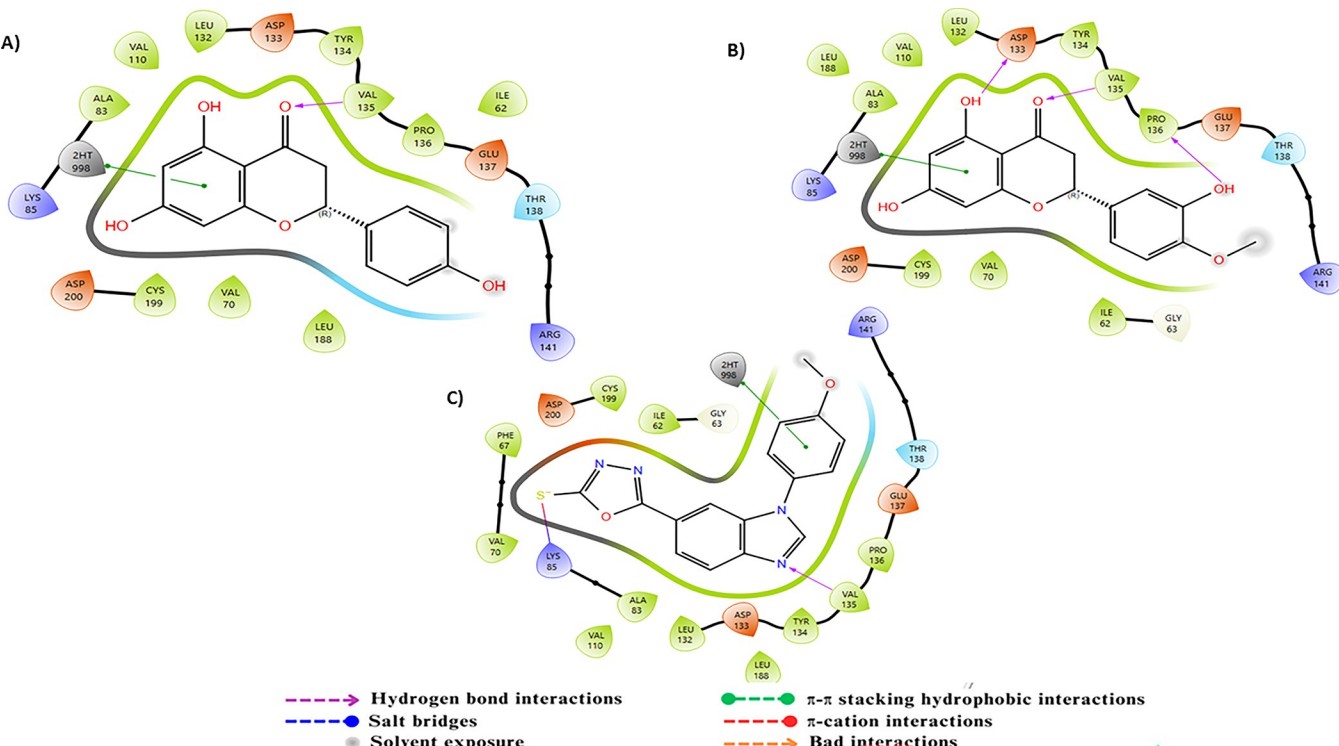

**Fig 7.** Two-dimensional (2D) representation of the docking poses in the active site of GSK-3β (PDB ID:3F88) for compounds A) C23, B) C24 and C) the reference.

these electronic parameters. Fig 8 illustrates the localized states of the HOMO-LUMO energies for the selected compounds, offering a clear visualization of areas with increased and decreased electron density within the molecules.

## Molecular dynamics simulations

To further support and validate our previous findings, we conducted a molecular dynamics study, a computational simulation technique used to examine the behaviour and stability of ligand-protein complexes under simulated physiological conditions by tracking the

**Table 4. Quantum parameters for selected compounds C23 and C24.**

| Compound | Quantum parameters | |
|---|---|---|
| | **C23** | **C24** |
| $E_{HOMO}$ (eV) | -6.285 | -5.988 |
| $E_{LUMO}$ (eV) | -1.660 | -1.602 |
| Egap (eV) | 4.626 | 4.385 |
| Electronegativity X | 3.972 | 3.795 |
| Hardness η | 2.313 | 2.193 |
| Softness σ | 0.432 | 0.456 |
| Chemical potential μ | -3.972 | -3.795 |
| Electrophilicity ω | 3.411 | 3.284 |
| Dipole moment (DEBYE) | 2.603 | 4.395 |
| Electronic energy | -955.203 | -1069.758 |

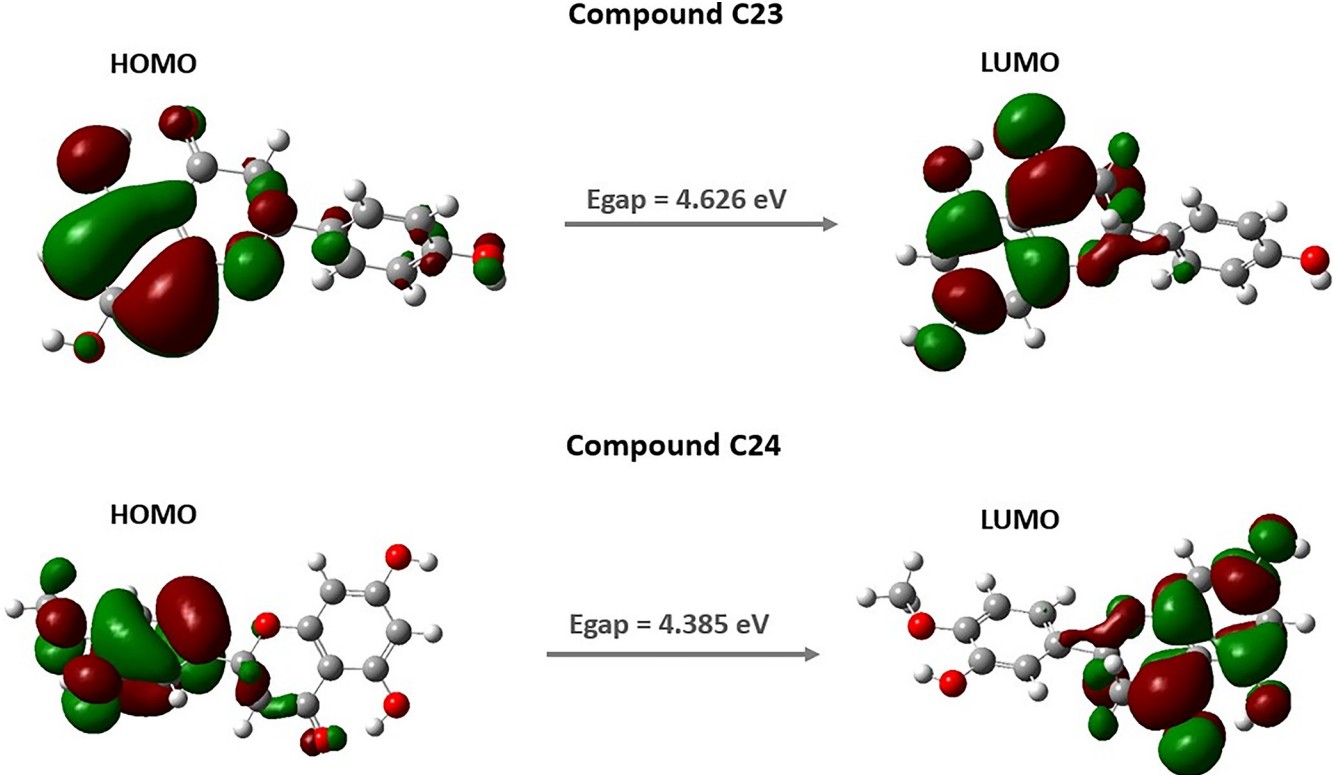

**Fig 8. Molecular orbital distribution plots of HOMO and LUMO of the selected compounds.**

movements of atoms and molecules. The Root Mean Square Deviation (RMSD) measures the average deviation of atomic positions from a reference structure over time; a low and stable RMSD indicates a stable conformation of the protein-ligand complex [77]. The Root Mean Square Fluctuation (RMSF) quantifies the average fluctuation of individual protein residues from their mean positions, with higher RMSF values indicating more flexible regions and lower values indicating more rigid regions. Protein-ligand interactions can involve hydrogen bonds, hydrophobic interactions, salt bridges, and other non-covalent forces, all of which are crucial for the stabilization of the complex and the efficacy of the ligand as an inhibitor [77]. In our study, we performed molecular dynamics simulations on the two selected compounds with the five targets AChE, BuChE, MAO-B, BACE-1, and GSK-3β to evaluate their behaviour and stability, and we assessed various parameters to gain insights into their potential as effective inhibitors.

For the AChE target, the molecular dynamics analysis of the protein-ligand complexes reveals notable stability (Fig 9). In the AChE-C23 complex, the protein RMSD shows some initial instability, with a gradual increase from 1.5 Å to approximately 1.9 Å towards the end of the simulation. This variation suggests moderate flexibility, although the overall protein structure remains relatively stable. Conversely, the ligand RMSD exhibits moderate fluctuations, ranging from 4.2 to 5.8 Å throughout the simulation, indicating a slightly less stable conformation. For the AChE-C24 complex, the protein RMSD starts at around 1.1 Å and remains relatively stable, with a slight increase to about 1.4 Å by the end of the simulation, reflecting good protein stability with minimal fluctuations. Similarly, the ligand RMSD shows little variation, with values between 4.2 and 5.6 Å, indicating a stable position of the ligand within the binding site. RMSF analysis reveals that certain regions of the proteins are more flexible. In the

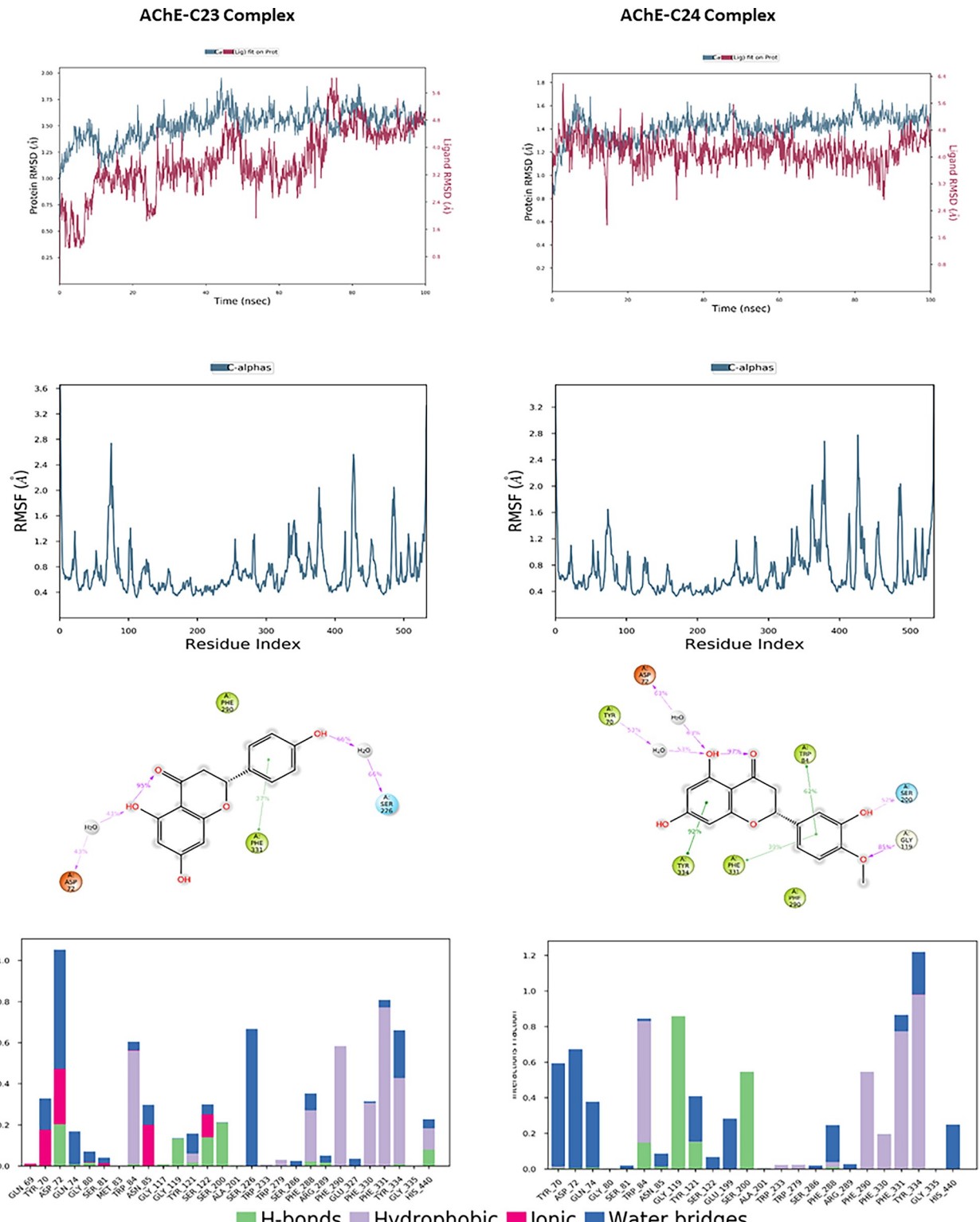

**Fig 9. The RMSD, RMSF plots and protein–ligand contacts of both complexes AChE-C23 and AChE-C24.**

AChE-C23 complex, RMSF peaks reach up to 2.8 Å, indicating increased flexibility in these specific regions, while the majority of residues fluctuate around 1 Å, suggesting relative rigidity. Similarly, for the AChE-C24 complex, RMSF peaks reach approximately 2.8 Å, indicating increased flexibility in some areas, while most residues also fluctuate around 1 Å, showing overall stability. Protein-ligand interactions revealed several significant hydrogen bonds and hydrophobic interactions for both compounds. For compound C23, residues such as Asp72 and Phe331 showed frequent interactions with interaction fractions of 1 and 0.8, respectively. For compound C24, residues Tyr334 and Phe331 exhibited significant interactions with interaction fractions of 1 and 0.6, respectively. These results suggest that both compounds are promising candidates for AChE inhibition, with a slight preference for C24 in terms of stability and frequency of critical interactions.

The molecular dynamics analysis of compounds C23 and C24 in interaction with BuChE revealed significant differences in terms of stability and flexibility, the results are illustrated in Fig 10. The RMSD of the protein in complex with C23 shows a slight progressive increase, reaching approximately 1.75 Å at the end of the simulation, indicating minor structural adjustments despite overall stability. The RMSD of ligand C23, on the other hand, begins around 1 Å, gradually increases to stabilize around 3 Å, then decreases to finally stabilize around 2 Å, reflecting more substantial structural changes over time and lesser stability. In comparison, the RMSD of the protein in complex with C24 stabilizes around 1.4 Å with some fluctuations, suggesting notable stability. The RMSD of ligand C24 begins at approximately 3 Å, stabilizes around 6 Å, and remains stable throughout the simulation, fluctuating slightly around 8 Å. The RMSF analysis of residues reveals greater flexibility in certain regions of the BuChE-C23 complex, with peaks reaching up to 3.5 Å, while the BuChE-C24 complex exhibits lower fluctuations, not exceeding 3 Å, indicating better structural stability. The 2D interaction diagrams show that C23 forms hydrogen bonds with Asp72, Ser226, and Trp82, as well as hydrophobic interactions with Phe331. In contrast, C24 interacts through hydrogen bonds with Tyr70, Asp72, Ser200, and Gly119, along with hydrophobic interactions with Trp84, Tyr334, and Phe331. The frequency of interactions highlights that C24 maintains more stable and consistent contacts with the residues of BuChE, corroborating the stability observed in the RMSD and RMSF analyses.

The molecular dynamics results for compounds C23 and C24 interacting with MAO-B are illustrated in Fig 11, revealing notable differences in stability and flexibility. For the MAO-B-C23 complex, the protein reaches a stable RMSD of approximately 2.4 to 3.0 Å after an initial adjustment phase, while the ligand shows relative stability with an RMSD fluctuating around 9.0 to 10.0 Å, suggesting a stable positioning of the ligand within the binding site. For the complex with C24, the protein RMSD starts at 1.5 Å and stabilizes around 2.4 Å, showing fluctuations throughout the simulation. The ligand RMSD begins to stabilize at around 4.2 Å, fluctuating throughout the simulation to approximately 3.6 Å, reflecting an initial instability followed by progressive stabilization, which is slightly more variable than C23. The RMSF analysis indicates relatively low residue fluctuations for C23, with notable peaks around specific residues reaching values close to 3.5 Å, indicating areas of increased flexibility. For C24, fluctuations are also concentrated around similar residues but are slightly higher, peaking at 4 Å, suggesting that certain regions of MAO-B are more flexible in the presence of C24. 2D interaction diagrams reveal that C23 forms hydrogen bonds with residues Ala263, Ile14, Ser15, Arg42, and Tyr60. C24 engages in hydrogen bonds with Arg42, Gly434, and Leu171, as well as hydrophobic interactions involving Phe343. The interaction frequency shows that C24 maintains more stable and consistent contacts with MAO-B residues, supporting a potentially more stable and stronger interaction.

Regarding BACE-1, Molecular dynamics simulations of the complexes formed between BACE-1 and the compounds C23 and C24 reveal notable differences in stability and flexibility, as illustrated in Fig 12. For the C23 complex, the protein achieves a stable RMSD around 1.5 to

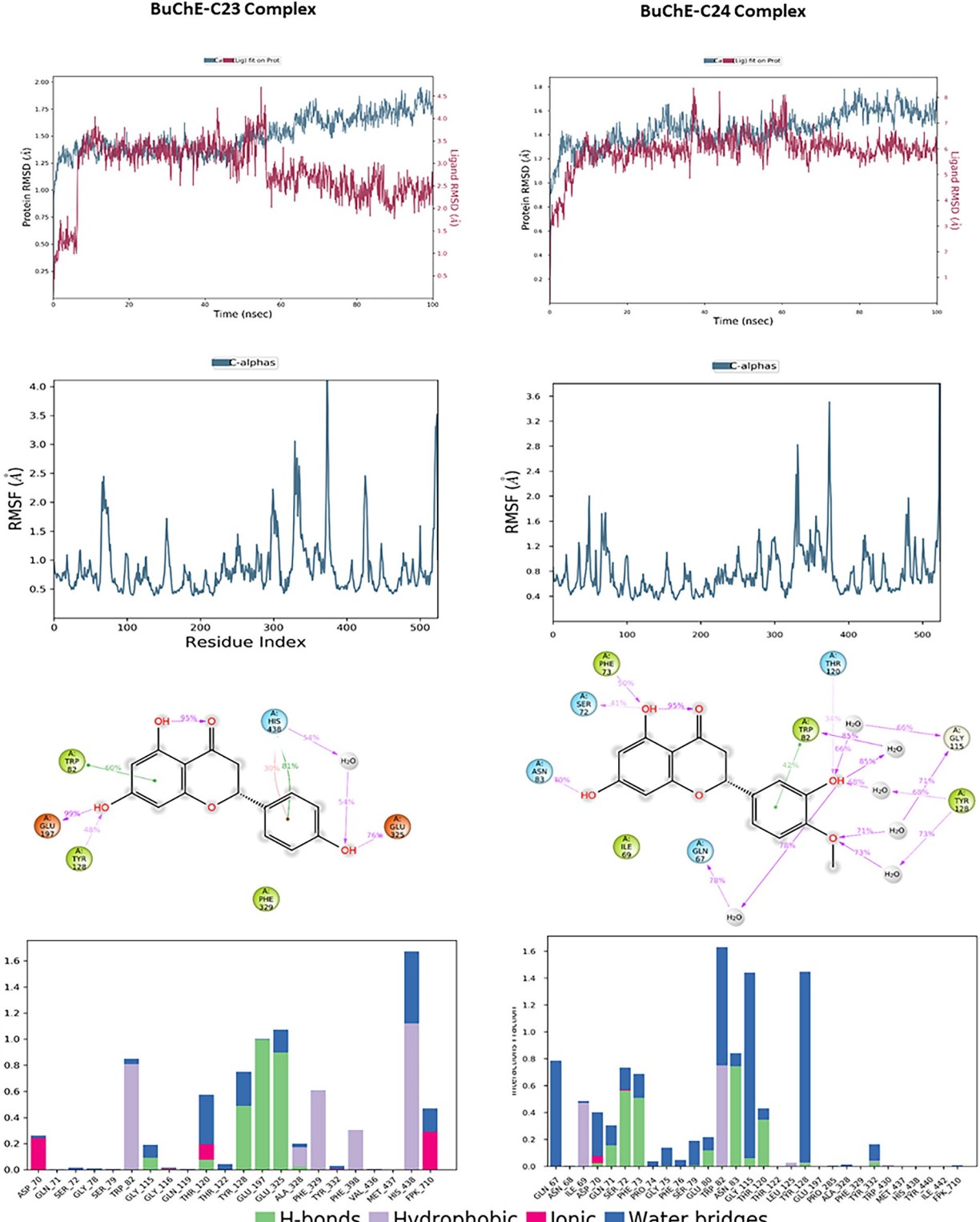

**Fig 10. The RMSD, RMSF plots and protein–ligand contacts of both complexes BuChE-C23 and BuChE-C24.**

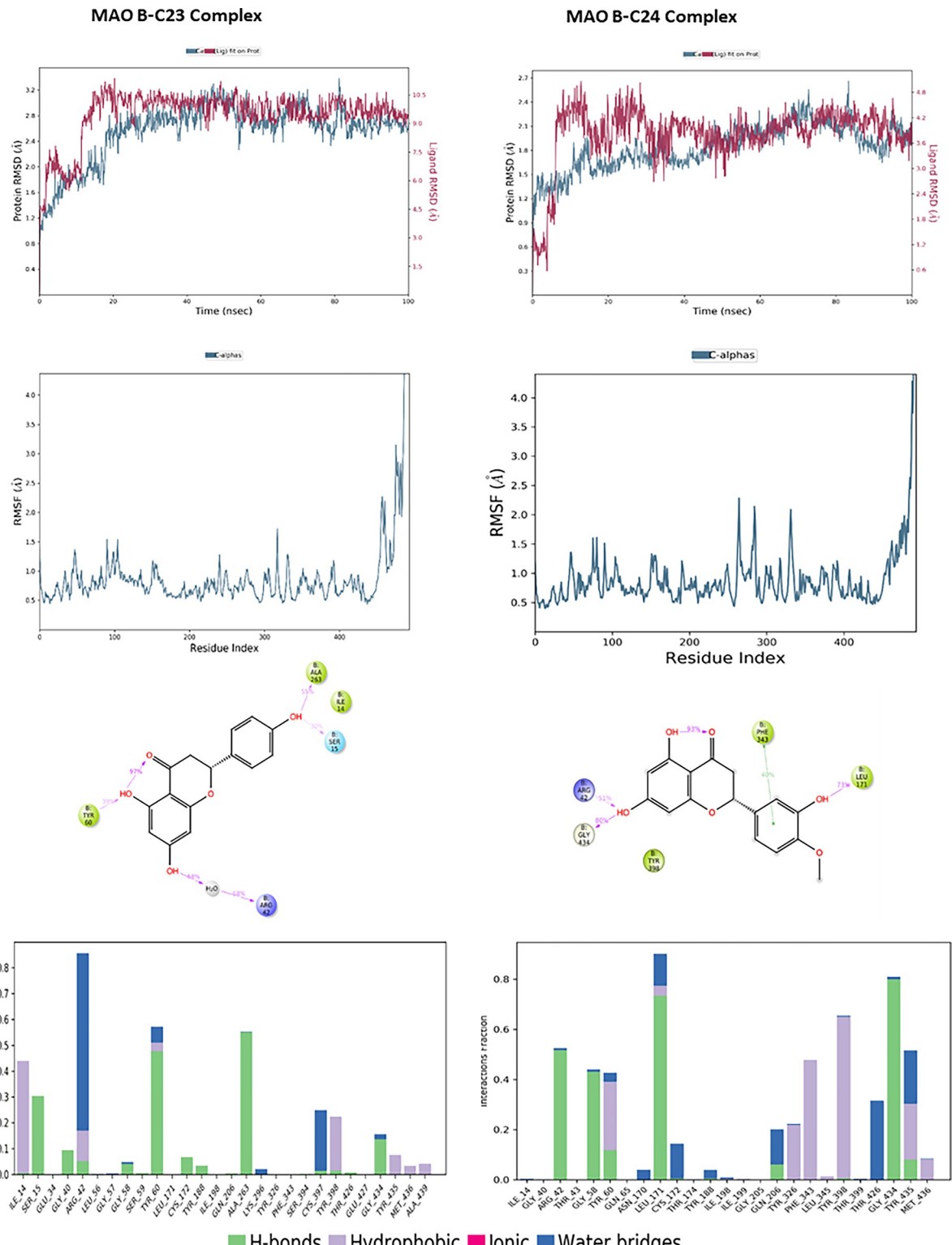

**Fig 11. The RMSD, RMSF plots and protein–ligand contacts of both complexes MAO-B-C23 and MAO-B-C24.**

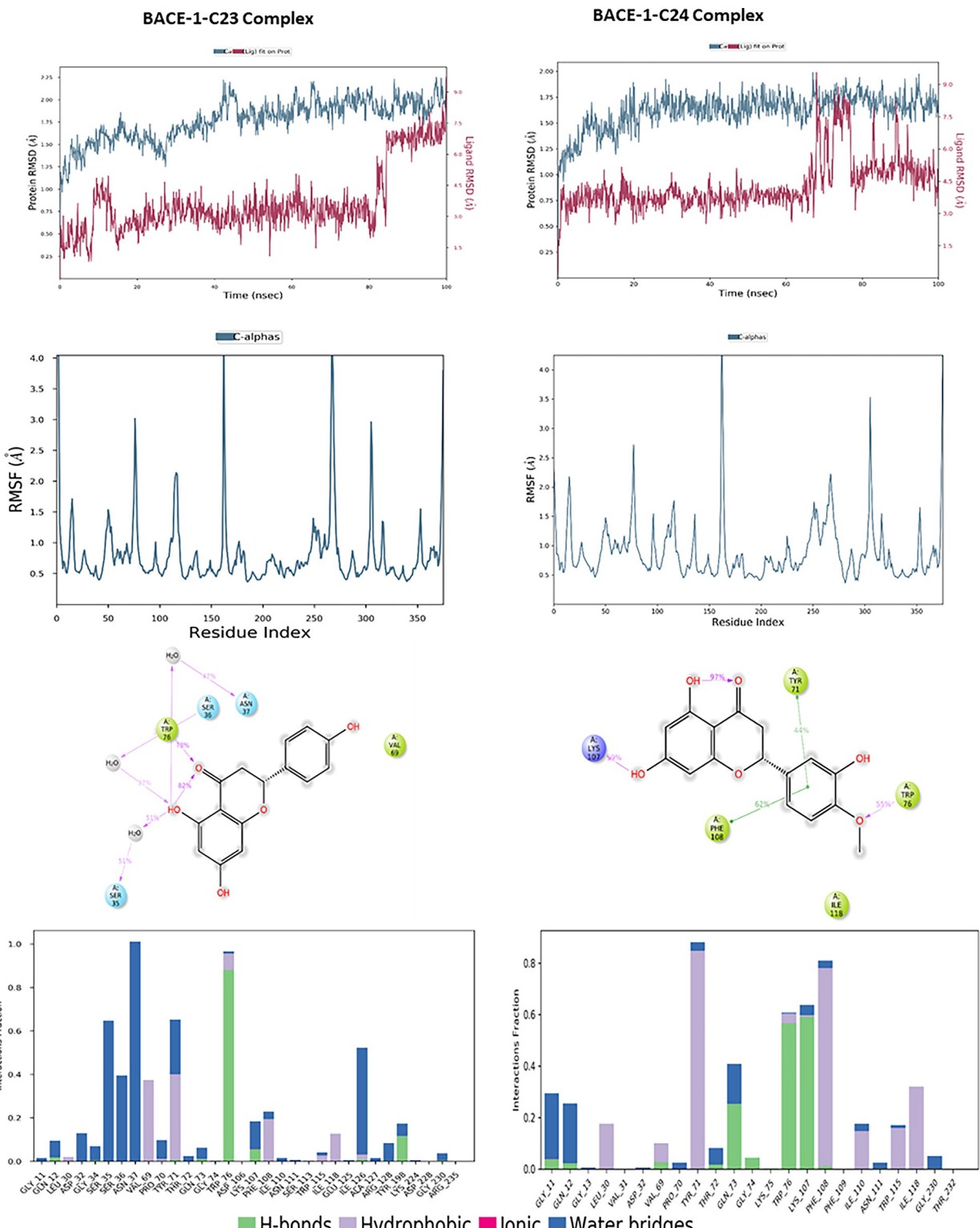

**Fig 12. The RMSD, RMSF plots and protein–ligand contacts of both complexes BACE-1-C23 and BACE-1-C24.**

2.0 Å throughout the simulation, indicating a relatively stable conformation, while the ligand shows good stability with an RMSD fluctuating between 2.5 and 4.5 Å, despite a slight reorganization towards the end of the simulation. In the case of the C24 complex, the protein exhibits similar stability but with a slightly lower RMSD, around 1.5 to 1.75 Å, suggesting a well-maintained conformation. The C24 ligand stabilizes around 4.5 Å up to 60 ns, after which it shows notable fluctuations, suggesting increased flexibility within the binding site. RMSF analysis reveals that BACE-1 residues in the presence of C23 experience greater fluctuations, reaching up to 4.0 Å for some residues, while C24 shows slightly lower fluctuations, below 3.5 Å, indicating marginally reduced flexibility. Protein-ligand interactions highlight hydrogen bonds with Trp76 and Ser35 for C23, whereas C24 forms hydrogen bonds with Lys107 and Trp76, as well as hydrophobic interactions with Phe108 and Tyr71. Interaction frequency analysis indicates that C24 maintains more stable and consistent contacts with critical BACE-1 residues.

For the target, GSK-3β, the molecular dynamics analysis of compounds C23 and C24 reveals significant differences in their stability and flexibility, the results are illustrated in Fig 13. The protein RMSD for compound C23 shows overall structural stability with fluctuations around 1.5 Å, occasionally reaching up to 2.5 Å, indicating a relatively stable conformation with some notable variations. The ligand RMSD for C23 fluctuates around 0.9 Å, suggesting a stable positioning of the ligand within the binding site. Similarly, for compound C24, the protein RMSD presents fluctuations around 1.5 Å, reaching approximately 2.0 Å, indicating structural stability comparable to that observed with C23. The ligand RMSD for C24 also fluctuates around 3 Å, indicating that the ligand remains stably anchored in the binding site. The RMSF for C23 reveals significant peaks at the indices of several residues, reaching up to 4 Å, indicating high flexibility in these specific regions, likely due to dynamic interactions between the protein and ligand. For C24, the RMSF also shows significant peaks at similar residue indices, reaching up to 4 Å, suggesting comparable flexible regions to those observed with C23. Protein-ligand interaction diagrams for C23 show that this compound forms multiple hydrogen bonds with key residues such as Asp133, Asp200, Lys85 and Val135, stabilizing the protein's conformation. For C24, interactions include bonds with residues like Asp133, Asp200 and Val135, also stabilizing the protein-ligand complex. Although both compounds exhibit nearly similar profiles of stability and flexibility, C24 shows greater fluctuation, which might indicate less consistent binding compared to C23. However, both compounds form strong interactions with key residues, suggesting they both have the potential to be effective inhibitors.

## MM-GBSA calculations

The MM-GBSA calculation was performed to evaluate the binding energy (ΔG_bind) of complexes formed by two selected compounds with five protein targets (AChE, BuChE, MAO-B, BACE-1, and GSK-3β). This method estimates the thermodynamic stability of protein-ligand complexes by accounting for the energetic contributions of molecular mechanics force fields and solvation effects [78]. The ΔG_bind values obtained, presented in the Table 5, indicate the strength of interactions between the ligands and their respective protein targets. A more negative ΔG_bind suggests stronger binding and a more favourable interaction between the ligand and the protein, indicating greater complex stability [79]. The results show that compound C23 exhibits favourable interactions with AChE (-45.62 kcal/mol), BuChE (-45.14 kcal/mol) and MAO-B (-45.77 kcal/mol). However, its interaction with BACE-1 is weaker (-19.22 kcal/mol), suggesting less favourable affinity, while the interaction with GSK-3β, (-40.97 kcal/mol) remains notable. In comparison, compound C24 shows even stronger interactions, particularly with AChE (-57.03 kcal/mol), MAO-B (-55.34 kcal/mol), and BACE-1 (-50.46 kcal/mol). These values indicate highly favourable binding and strong interactions with these targets.

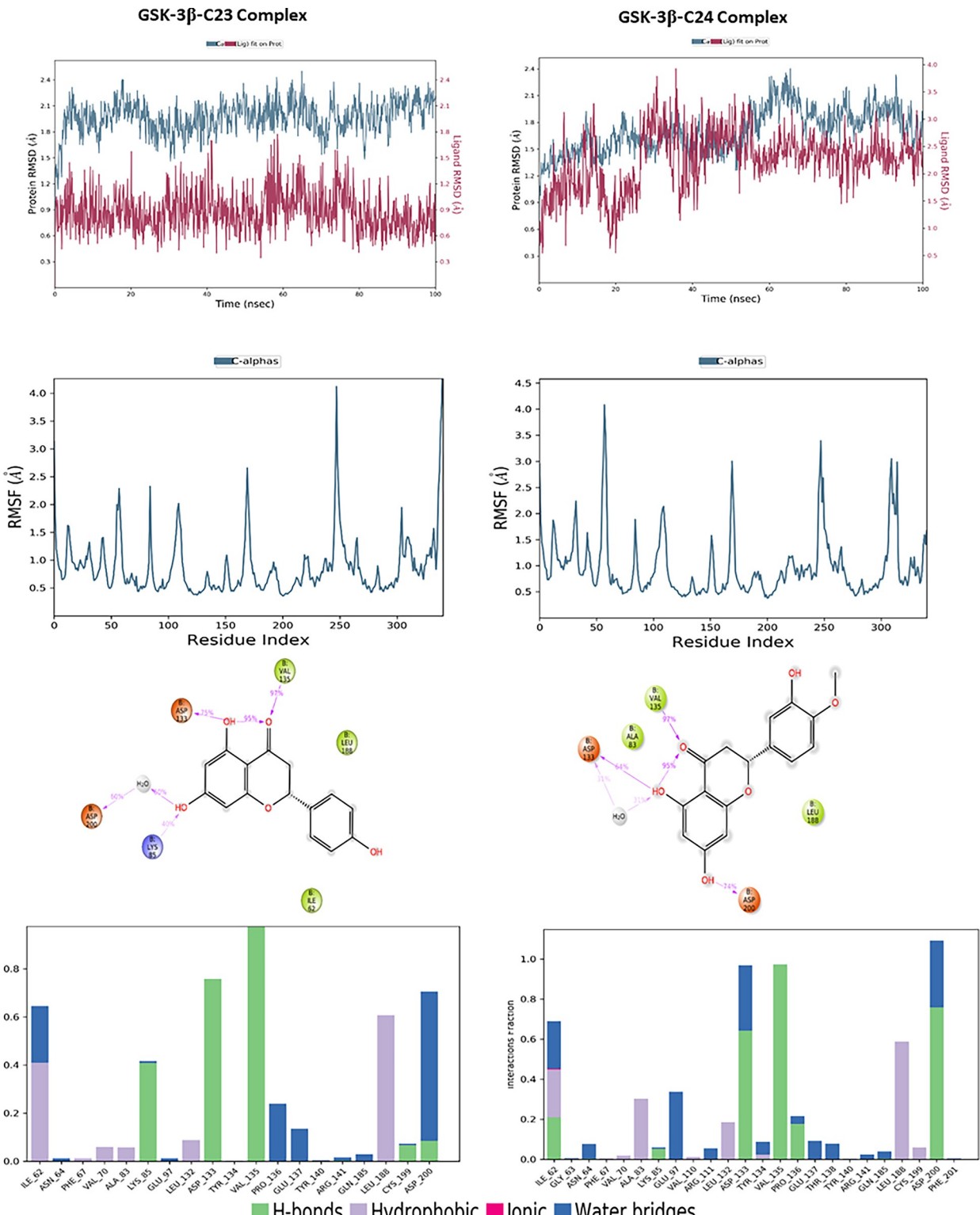

**Fig 13. The RMSD, RMSF plots and protein–ligand contacts of both complexes GSK-3β-C23 and GSK-3β-C24.**

**Table 5. MM-GBSA binding affinity results for the selected compounds.**

| Compound | MM-GBSA binding affinities (kcal/mol) | | | | |
|---|---|---|---|---|---|
| | AChE | BuChE | MAO-B | BACE-1 | GSK-3β |
| C23 | -45.62 | -45.14 | -45.77 | -19.22 | -40.97 |
| C24 | -57.03 | -42.25 | -55.34 | -50.46 | -52.71 |

Additionally, C24 also exhibits significant binding energies with BuChE (-42.25 kcal/mol) and GSK-3β (-52.71 kcal/mol). Both selected compounds, C23 and C24, demonstrate significant negative binding energies with the five studied targets, indicating favourable interactions and potentially stable complexes. However, C24 appears to have overall stronger and more favourable interactions compared to C23, suggesting that C24 could be a more promising candidate.

## Conclusion

This study highlights the significant potential of phytochemical compounds from Moroccan medicinal plants as multi-target candidates for the treatment of Alzheimer's disease through computational approaches. Through rigorous analyses, ranging from the initial screening via molecular docking of a database of 386 phytochemical compounds and the evaluation of drug-likeness and ADMET parameters to molecular dynamics and DFT studies, we identified two potential lead compounds: naringenin (C23) and hesperetin (C24). These compounds, derived from the plant *Anabasis aretioides*, abundant in the Errachidia region of Morocco, demonstrated favourable pharmacokinetic profiles. They proved to be effective multi-target agents for AChE, BuChE, MAO-B, BACE-1, and GSK-3β, due to their significant binding affinities for these five targets. The results of DFT and molecular dynamics studies confirmed their stability, suggesting that these two compounds are strong multi-target candidates for inhibiting the studied targets, with a slight preference for compound C24 in terms of stability and frequency of critical interactions. In the future, our research will focus on rigorous experimental investigations, including *in vitro* and *in vivo* studies, to elucidate the potential efficacy and mechanisms of action of these compounds against AD. By combining computational data with thorough experimental validation, we aim to significantly advance the field of anti-Alzheimer drug discovery.

## Supporting information

**S1 Table. 2D structures and detailed information on the studied compounds.** (DOCX)

## Author Contributions

**Conceptualization:** Fatima Zahra Guerguer, Amal Bouribab, El Mehdi Karim, Meriem Khedraoui.

**Data curation:** Fatima Zahra Guerguer, Amal Bouribab.

**Formal analysis:** Fatima Zahra Guerguer, Amal Bouribab.

**Funding acquisition:** Yasir S. Raouf, Abdelouahid Samadi.

**Investigation:** Samir Chtita.

**Methodology:** Samir Chtita.

**Project administration:** Samir Chtita.

**Resources:** Samir Chtita.

**Software:** Samir Chtita.

**Supervision:** Samir Chtita.

**Validation:** Yasir S. Raouf, Abdelouahid Samadi, Samir Chtita.

**Visualization:** Fatiha Amegrissi, Yasir S. Raouf, Abdelouahid Samadi, Samir Chtita.

**Writing – original draft:** Fatima Zahra Guerguer, Amal Bouribab.

**Writing – review & editing:** Fatima Zahra Guerguer, El Mehdi Karim, Meriem Khedraoui, Fatiha Amegrissi.

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
