## [Decision Letter · Decision Letter 0]

11 Sep 2024

PONE-D-24-36446Moroccan natural products for multitarget-based treatment of Alzheimer’s disease: A computational study involving molecular docking, ADMET analysis, density functional theory, and molecular dynamics simulationsPLOS ONE

Dear Dr. Chtita,

Thank you for submitting your manuscript to PLOS ONE. After careful consideration, we feel that it has merit but does not fully meet PLOS ONE’s publication criteria as it currently stands. Therefore, we invite you to submit a revised version of the manuscript that addresses the points raised during the review process.

We look forward to receiving your revised manuscript.

Kind regards,

Prashant Singh

Academic Editor

PLOS ONE

2. Please note that PLOS ONE has spec6ific guidelines on code sharing for submissions in which author-generated code underpins the findings in the manuscript. In these cases, all author-generated code must be made available without restrictions upon publication of the work. Please review our guidelines at https://journals.plos.org/plosone/s/materials-and-software-sharing#loc-sharing-code and ensure that your code is shared in a way that follows best practice and facilitates reproducibility and reuse.

4. Thank you for stating the following financial disclosure: “AS and YSR acknowledge the support of the UAEU through an internal Start-up grant 2023 (Grant Code G00004400) and an internal Start-up grant 2024 (Grant 12S156), respectively.”

5. We note that your Data Availability Statement is currently as follows: All relevant data are within the manuscript and its Supporting Information files.

Please confirm at this time whether or not your submission contains all raw data required to replicate the results of your study. Authors must share the “minimal data set” for their submission. PLOS defines the minimal data set to consist of the data required to replicate all study findings reported in the article, as well as related metadata and methods (https://journals.plos.org/plosone/s/data-availability#loc-minimal-data-set-definition). For example, authors should submit the following data: - The values behind the means, standard deviations and other measures reported; - The values used to build graphs; - The points extracted from images for analysis. Authors do not need to submit their entire data set if only a portion of the data was used in the reported study. If your submission does not contain these data, please either upload them as Supporting Information files or deposit them to a stable, public repository and provide us with the relevant URLs, DOIs, or accession numbers. For a list of recommended repositories, please see https://journals.plos.org/plosone/s/recommended-repositories. If there are ethical or legal restrictions on sharing a de-identified data set, please explain them in detail (e.g., data contain potentially sensitive information, data are owned by a third-party organization, etc.) and who has imposed them (e.g., an ethics committee). Please also provide contact information for a data access committee, ethics committee, or other institutional body to which data requests may be sent. If data are owned by a third party, please indicate how others may request data access.

Reviewers' comments:

Reviewer's Responses to Questions

**Comments to the Author**

1. Is the manuscript technically sound, and do the data support the conclusions?

Reviewer #1: Yes

Reviewer #2: Partly

Reviewer #3: Yes

2. Has the statistical analysis been performed appropriately and rigorously? 

Reviewer #1: N/A

Reviewer #2: Yes

Reviewer #3: Yes

3. Have the authors made all data underlying the findings in their manuscript fully available?

Reviewer #1: Yes

Reviewer #2: Yes

Reviewer #3: Yes

4. Is the manuscript presented in an intelligible fashion and written in standard English?

Reviewer #1: Yes

Reviewer #2: Yes

Reviewer #3: No

5. Review Comments to the Author

Reviewer #1: The manuscript deals with a pertinent and timely topic, which deserves continued relevance until further developments emerge in the fight against Alzheimer's. In terms of the protocol and computational methods used, the study was very thorough and careful. The authors also took care to explain all the analyses and tools used in a very didactic and detailed way.

I'll just point out a few details:

1. "55 million people were living with dementia in 2019..."; I think the authors can have even more up-to-date values, from 2023 in WHO.

2. Protein Preparation:

a) What were the criteria for choosing the PDB codes, the resolution, the year, or the natural ligand?

b) Protein Preparation Wizard tool... please add, within Schrodinger's package...

3. Ligand Preparation:

What was the point of generating 32 conformers of each of the 386 phytochemicals, at different pHs? After this step, was a more representative conformer chosen, or one with lower energy? Or were they all subjected to docking? It is not clear...

4. "6-311G (d, p) quantum database" should be 'basis set' not database.

5. Table 1: It would be interesting to have the 2D structures of the 17 candidates, and their extract (similar to the SI), for a more direct comparison between the structures with the best binding.

6. Figures 9-13: What is the meaning of the colour scheme on the bottom plots (interactions fraction bars)?

Reviewer #2: The manuscript titled "Moroccan natural products for multitarget-based treatment of Alzheimer’s disease: A computational study involving molecular docking, ADMET analysis, density functional theory, and molecular dynamics simulations". Overall, this manuscript presents valuable research with well-executed experiments and detailed analysis. Addressing the below points will enhance the clarity and impact of the manuscript

1. Abstract: The abstract provides a solid overview but can be more concise. Consider revising the ADMET and MD simulation explanations to simplify the language, especially since these are technical terms that might distract from the main message. Specify what "promising results" from MD and DFT entail.

2. Introduction

Include Specific Rationale for Choosing Moroccan Plants: While the manuscript mentions Morocco’s rich biodiversity, further justification for why these specific phytochemicals were chosen would be beneficial. For example, do these plants have a historical use in neurodegenerative diseases?

3. Material and Methods:

Details on Dataset Selection: There is a brief mention of 386 compounds, but the criteria for selecting these compounds should be elaborated. Were these chosen based on previous evidence of activity against AD? Provide more details on why these 17 compounds were shortlisted.

Docking Methodology: The docking process is well described. However, the text mentions that Glide XP was used but does not discuss why this level of precision was necessary or beneficial for your study.

MD Simulations: The protein preparation process (e.g., removal of water molecules, addition of hydrogen atoms) is well described. However, explain why a cutoff of 0.5 Å for water molecule removal was chosen, as this could affect docking accuracy.

Molecular Quantum Analysis: The DFT analysis is informative, but more justification for selecting the B3LYP functional and 6-311G(d,p) basis set is needed.

4. Results and Discussion:

• Compare the ADMET profiles of C23 and C24 with existing AD drugs.

• The MD simulations are well executed, but more justification is needed for selecting a 100 ns simulation time. Did shorter or longer simulations provide different insights into the stability of ligand-protein interactions?

• The MD simulation results are discussed thoroughly but, it may be helpful to give a brief explanation of why RMSD/RMSF analyses are crucial for understanding ligand-protein stability.

5. Consistency in Terminology: Be consistent with the use of technical terms, such as "multi-target-directed ligands" (MTDL) and "natural compounds." Some sections refer to ligands, while others switch to phytochemicals, so standardizing this will enhance clarity.

6. Sentence structure: Some sentences are complex and difficult to follow, particularly in the methodology and results sections. Shortening and simplifying some of these would enhance readability.

Reviewer #3: Moroccan natural products for multitarget-based treatment of Alzheimer’s disease: A computational study involving molecular docking, ADMET analysis, density functional theory, and molecular dynamics simulations

The title is too long! The authors should consider removing everything that appears after the word ‘study’.

The manuscript is well written but there are some grammar errors that can be fixed by a proof-reader. There are also many phrases/words that are redundant. For example, this sentence ‘Regarding distribution, achieving effective drug penetration through the blood-brain barrier (BBB) is challenging, particularly for treating central nervous system (CNS) disorders due to tightly-packed endothelial cells, which are practically impregnable.’ Authors did not number the pages hence I can’t refer you to the page! I am not sure what information the authors want to convey. Are you referring to distribution of the drug or achieving drug penetration?

There are a many other similar grammar errors in the manuscript that need to be fixed to avoid ambiguity.

Reference to the figures and tables in the text: the author has used lower case in cases when referring to figures and tables. Please check and be consistent. For example: Table 1 should appear as Table 1 and NOT table 1!

The figures that depict the docking poses are not very clear. It is not possible to see the text in the figures. Either increase the size and show the images A, B, C one below the other. Or improve the clarify of the image.

The figure captions can be corrected as well. For example it figure captions should read as follows: Figure 3 : Two-dimensional (2D) representation of the docking poses in the active site of AChE (PDB ID:1EVE) for compounds A) C23, B) C24 and C) the reference

Figures 9-13 that has the structures showing the docking, are also not very clear. Again, improve the clarity of these images.

6. PLOS authors have the option to publish the peer review history of their article (what does this mean?). If published, this will include your full peer review and any attached files.

Reviewer #1: No

Reviewer #2: No

Reviewer #3: No

---

## [Author Response · Author response to Decision Letter 0]

16 Oct 2024

Manuscript (ID: PONE-D-24-36446)

Moroccan natural products for multitarget-based treatment of Alzheimer’s disease: A computational study

Fatima Zahra Guerguer (1), Amal Bouribab (1), El Mehdi Karim (1), Meriem Khedraoui (1), Fatiha Amegrissi (1), Yasir S. Raouf (2), Abdelouahid Samadi (2, *), and Samir Chtita (1, *)

Dear Editor,

On behalf of our research team, we would like to express our sincere gratitude to you and the reviewers for the time and effort devoted to evaluating our manuscript submitted to PLOS ONE. We greatly appreciate the constructive comments and suggestions, which have undoubtedly contributed to improving the quality of our work.

We have carefully considered all the feedback and have made the necessary revisions to the manuscript accordingly. Each comment has been thoroughly addressed, and our responses are provided below. The changes made to the manuscript are highlighted in yellow. 

We hope that these revisions meet the expectations and high standards of PLOS ONE. Once again, we thank you for this opportunity to contribute to your journal and remain open to any further suggestions or recommendations.

We look forward to your response and extend our best regards.

Sincerely,

Pr. Samir Chtita

Response to reviewer’s comments

Reviewer Comments:

REVIEWER 1: 

Comments: The manuscript deals with a pertinent and timely topic, which deserves continued relevance until further developments emerge in the fight against Alzheimer's. In terms of the protocol and computational methods used, the study was very thorough and careful. The authors also took care to explain all the analyses and tools used in a very didactic and detailed way.

I'll just point out a few details:

1. "55 million people were living with dementia in 2019..."; I think the authors can have even more up-to-date values, from 2023 in WHO.

Thank you for your valuable suggestion. We have updated the revised manuscript with the most recent values from the World Health Organization (WHO) as of 2023.

2. Protein Preparation:

a) What were the criteria for choosing the PDB codes, the resolution, the year, or the natural ligand?

We sincerely thank you for your thoughtful and constructive comments.

Regarding the selection of PDB codes, our approach was guided by three main criteria. Firstly, we prioritized structures with a crystallographic resolution below 2.6 Å to ensure high accuracy in the analysis of molecular interactions. Secondly, we considered the publication date to ensure that the data used were both current and relevant. Lastly, the presence of appropriate co-ligands played a key role in our selection. For example, for acetylcholinesterase (AChE), we specifically chose a structure co-crystallized with the reference drug. For other targets such as BuChE, MAO-B, BACE-1, and GSK-3, we opted for PDB codes featuring co-ligands that are extensively cited and recognized in the scientific literature, ensuring the biological significance of the complexes studied.

We hope this explanation clarifies our selection process, and we remain at your disposal for any further questions or suggestions.

b) Protein Preparation Wizard tool... please add, within Schrodinger's package...

Thank you for your valuable suggestion. The Protein Preparation Wizard tool from the Schrödinger suite was indeed used for the preparation of the protein structures. Following your recommendation, we have now indicated this in the section dedicated to protein preparation in the revised version of the manuscript.

3. Ligand Preparation:

a) What was the point of generating 32 conformers of each of the 386 phytochemicals, at different pHs? After this step, was a more representative conformer chosen, or one with lower energy? Or were they all subjected to docking? It is not clear...

Thank you for your insightful question regarding the generation of conformers for the phytochemicals. We employed Schrödinger's LigPrep tool to generate 32 conformers for each of the 386 phytochemical compounds to effectively capture their conformational diversity across different pH levels. Following this step, all conformers underwent virtual screening through a docking process. For each compound, the best pose was selected based on the lowest docking score, which indicates the most stable interaction with the target.

4. "6-311G (d, p) quantum database" should be 'basis set' not database.

Thank you for your valuable observation. we have corrected "6-311G (d, p) quantum database" to "basis set" in the revised manuscript.

5. Table 1: It would be interesting to have the 2D structures of the 17 candidates, and their extract (similar to the SI), for a more direct comparison between the structures with the best binding.

Thank you for your insightful suggestion regarding Table 1. We have modified the table to include the 2D structures of the 17 candidates, along with their extracts, in the revised manuscript. This addition should facilitate a more direct comparison of the structures with the best binding properties. We appreciate your valuable feedback.

6. Figures 9-13: What is the meaning of the color scheme on the bottom plots (interactions fraction bars)?

Thank you for your question regarding the color scheme used in the interaction fraction bars of Figures 9-13. We have added a legend to these figures in the revised manuscript to clarify the meaning of the color scheme. We appreciate your attention to this detail.

REVIEWER 2: 

Comments: The manuscript titled "Moroccan natural products for multitarget-based treatment of Alzheimer’s disease: A computational study involving molecular docking, ADMET analysis, density functional theory, and molecular dynamics simulations". Overall, this manuscript presents valuable research with well-executed experiments and detailed analysis. Addressing the below points will enhance the clarity and impact of the manuscript.

1. Abstract: The abstract provides a solid overview but can be more concise. Consider revising the ADMET and MD simulation explanations to simplify the language, especially since these are technical terms that might distract from the main message. Specify what "promising results" from MD and DFT entail.

Thank you for your valuable feedback on the abstract. We have made the necessary revisions in the revised manuscript to enhance its clarity and conciseness. We hope these adjustments meet your expectations.

2. Introduction

Include Specific Rationale for Choosing Moroccan Plants: While the manuscript mentions Morocco’s rich biodiversity, further justification for why these specific phytochemicals were chosen would be beneficial. For example, do these plants have a historical use in neurodegenerative diseases?

Thank you for your insightful suggestion regarding the rationale for choosing Moroccan plants. We have addressed this point and provided further justification for the selection of these specific phytochemicals in the revised manuscript's introduction.

3. Material and Methods:

a) Details on Dataset Selection: There is a brief mention of 386 compounds, but the criteria for selecting these compounds should be elaborated. Were these chosen based on previous evidence of activity against AD? Provide more details on why these 17 compounds were shortlisted.

Thank you for your insightful comments regarding the criteria for selecting the dataset. In the revised manuscript, we have addressed this concern in the introduction. The database we utilized comprises endemic plants from Morocco, totaling 386 phytochemical compounds. These plants are widely recognized in traditional medicine for their therapeutic properties against a variety of diseases, including neurodegenerative disorders, cancer, and diabetes, with particular emphasis on their neuroprotective, antioxidant, and anti-inflammatory activities. Our research aims to highlight and investigate the potential of these compounds as multi-target candidates for Alzheimer’s disease treatment. We believe that these phytochemicals offer promising therapeutic avenues by simultaneously targeting multiple pathological mechanisms involved in the disease.

We trust that this justifies our selection of compounds. We remain at your disposal for any further questions or suggestions.

b) Docking Methodology: The docking process is well described. However, the text mentions that Glide XP was used but does not discuss why this level of precision was necessary or beneficial for your study.

Thank you for your valuable feedback regarding the docking methodology. In the revised manuscript, we have elaborated on the use of Glide XP and its significance in our study.

c) MD Simulations: The protein preparation process (e.g., removal of water molecules, addition of hydrogen atoms) is well described. However, explain why a cutoff of 0.5 Å for water molecule removal was chosen, as this could affect docking accuracy.

We greatly appreciate your valuable feedback regarding the molecular dynamics simulations. In the revised manuscript, we have elaborated on the rationale for selecting a cutoff of 0.5 Å for the removal of water molecules.

d) Molecular Quantum Analysis: The DFT analysis is informative, but more justification for selecting the B3LYP functional and 6-311G(d,p) basis set is needed.

We appreciate your valuable feedback regarding the molecular quantum analysis. In the revised manuscript, we have elaborated on the rationale behind selecting the B3LYP functional and the 6-311G(d,p) basis set.

4. Results and Discussion:

a) Compare the ADMET profiles of C23 and C24 with existing AD drugs.

Thank you for your constructive feedback regarding the comparison of ADMET profiles of compounds C23 and C24 with existing Alzheimer’s drugs. In the revised version of the manuscript, we have included the pharmacokinetic data for donepezil, a reference drug, to enable a more detailed comparison. This addition allows for a better assessment of the performance of the studied compounds in relation to a commonly used treatment.

b) The MD simulations are well executed, but more justification is needed for selecting a 100 ns simulation time. Did shorter or longer simulations provide different insights into the stability of ligand-protein interactions?

We sincerely appreciate your insightful comment regarding the duration of the molecular dynamics simulations. In the revised manuscript, we have provided a thorough justification for selecting a simulation time of 100 ns. This duration is adequate to sufficiently capture complex molecular behaviors while minimizing the risk of overlooking critical dynamic events that may occur during shorter simulations.

We acknowledge that extending the simulation duration could yield additional insights and enrich our analysis. However, we must also ensure that we establish a sound compromise between accuracy and computational cost. In this study, several simulations were conducted, which enabled us to obtain robust results while adhering to computational time constraints. We are open to exploring longer-duration simulations in future studies to further enhance our understanding of molecular interactions.

c) The MD simulation results are discussed thoroughly but, it may be helpful to give a brief explanation of why RMSD/RMSF analyses are crucial for understanding ligand-protein stability.

We appreciate your valuable feedback regarding the discussion of the MD simulation results. In the revised manuscript, we have elaborated on the significance and insights provided by both the RMSD and RMSF analyses in understanding ligand-protein stability. These metrics are essential for assessing the conformational stability of the protein-ligand complex and evaluating the flexibility of the protein and ligand during the simulations. This detailed explanation has been incorporated into the results section of the dynamics analysis to enhance clarity and provide a comprehensive understanding.

d) Consistency in Terminology: Be consistent with the use of technical terms, such as "multi-target-directed ligands" (MTDL) and "natural compounds." Some sections refer to ligands, while others switch to phytochemicals, so standardizing this will enhance clarity.

We sincerely appreciate your insightful comment regarding the consistency in terminology within the manuscript. We have indeed ensured that the use of terms is standardized in the revised version. We value your suggestion as it helps enhance the clarity of the text. We remain at your disposal for any further questions or suggestions.

e) Sentence structure: Some sentences are complex and difficult to follow, particularly in the methodology and results sections. Shortening and simplifying some of these would enhance readability.

Thank you for your feedback. In the revised manuscript, we have made an effort to simplify and enhance the clarity of complex sentences to improve readability. We appreciate your valuable suggestions.

REVIEWER 3: 

Comments: Moroccan natural products for multitarget-based treatment of Alzheimer’s disease: A computational study involving molecular docking, ADMET analysis, density functional theory, and molecular dynamics simulations.

1. The title is too long! The authors should consider removing everything that appears after the word ‘study’.

Thank you for your comment. In line with your suggestion, we have shortened the title to make it more concise and clearer.

2. The manuscript is well written but there are some grammar errors that can be fixed by a proof-reader. There are also many phrases/words that are redundant. For example, this sentence ‘Regarding distribution, achieving effective drug penetration through the blood-brain barrier (BBB) is challenging, particularly for treating central nervous system (CNS) disorders due to tightly-packed endothelial cells, which are practically impregnable.’ Authors did not number the pages hence I can’t refer you to the page! I am not sure what information the authors want to convey. Are you referring to distribution of the drug or achieving drug penetration?

Thank you for this constructive feedback. The sentence you highlighted is intended to address the issue of drug distribution, particularly its ability to cross the blood-brain barrier (BBB). As part of the ADMET properties, distribution includes the evaluation of BBB penetration, which is crucial for treating central nervous system (CNS) disorders.

In the manuscript, we structured the discussion to sequentially analyze each ADMET property, distinguishing between Absorption, Distribution, Metabolism, Excretion, and Toxicity, to better interpret the parameters associated with each. However, we understand that some sentences may appear redundant or unclear. As a result, we have revised the writing to eliminate these redundancies and clarify the message. 

3. There are a many other similar grammar errors in the manuscript that need to be fixed to avoid ambiguity.

Thank you for your valuable feedback regarding the grammatical errors present in the manuscript. We have made significant efforts to address these issues and enhance the clarity of the text. We have carefully reviewed the document to correct grammatical mistakes and eliminate ambiguities. Your insights are greatly appreciated.

4. Reference to the figures and tables in the text: the author has used lower case in cases when referring to figures and tables. Please check and be consistent. For example: Table 1 should appear as Table 1 and NOT table 1!

Thank you for your observation. We have corrected the references to ensure consistency throughout the text. We appreciate your attention to detail, and these revisions have been implemented in the revised manuscript.

5. The figures that depict the docking poses are not very clear. It is not possible to see the text in the figures. Either increase the size and show the images A, B, C one below the other. Or improve the clarify of the image.

Thank you for your feedback regarding the clarity of the docking pose figures. We have made improvements to enhance the quality of the images. We appreciate your suggestions.

6. The figure captions can be corrected as well. For example it figure captions should read as follows: Figure 3: Two-dimensional (2D) representation of the docking poses in the active site of AChE (PDB ID:1EV

---

## [Decision Letter · Decision Letter 1]

24 Oct 2024

Moroccan natural products for multitarget-based treatment of Alzheimer’s disease: A computational study involving molecular docking, ADMET analysis, density functional theory, and molecular dynamics simulations

PONE-D-24-36446R1

Dear Dr. Chtita,

We’re pleased to inform you that your manuscript has been judged scientifically suitable for publication and will be formally accepted for publication once it meets all outstanding technical requirements.

Kind regards,

Prashant Singh

Academic Editor

PLOS ONE

Additional Editor Comments (optional):

I am pleased to inform you that your submission has been favorably reviewed and accepted for publication.

Reviewers' comments:

Reviewer's Responses to Questions

**Comments to the Author**

1. If the authors have adequately addressed your comments raised in a previous round of review and you feel that this manuscript is now acceptable for publication, you may indicate that here to bypass the “Comments to the Author” section, enter your conflict of interest statement in the “Confidential to Editor” section, and submit your "Accept" recommendation.

Reviewer #1: All comments have been addressed

Reviewer #2: All comments have been addressed

Reviewer #3: All comments have been addressed

2. Is the manuscript technically sound, and do the data support the conclusions?

Reviewer #1: Yes

Reviewer #2: Yes

Reviewer #3: Yes

3. Has the statistical analysis been performed appropriately and rigorously? 

Reviewer #1: N/A

Reviewer #2: Yes

Reviewer #3: N/A

4. Have the authors made all data underlying the findings in their manuscript fully available?

Reviewer #1: Yes

Reviewer #2: Yes

Reviewer #3: Yes

5. Is the manuscript presented in an intelligible fashion and written in standard English?

Reviewer #1: Yes

Reviewer #2: Yes

Reviewer #3: Yes

6. Review Comments to the Author

Reviewer #1: The authors have responded to my comments, but I would like to see questions 2 and 3 also explained clearly in the body of the article, not only in reply to my comments (2. What were the criteria for choosing the PDB codes, the resolution, the year, or the natural ligand? and 3. a)What was the aim of generating 32 conformers for each of the 386 phytochemicals, at different pHs? After this stage, was a more representative conformer chosen, or one with lower energy? Or were they all subjected to docking?).

Figures 9 and 10 could be improved: resolution, type font and font size on the axes, and legend boxes. Please make them uniform and clear.

Reviewer #2: The authors have thoroughly revised the manuscript in accordance with the provided comment. The revisions effectively address the comments and improve the clarity and overall quality of the manuscript. The experimental methods, computational studies, and discussion of the results are now presented in a clear and concise manner, making the research contributions more evident. Based on the revisions made, the manuscript is now ready for publication.

Reviewer #3: All comments and corrections that were requested, have been addressed by the authors. The manuscript is acceptable.

7. PLOS authors have the option to publish the peer review history of their article (what does this mean?). If published, this will include your full peer review and any attached files.

Reviewer #1: No

Reviewer #2: No

Reviewer #3: No

---

## [Editor Report · Acceptance letter]

2 Dec 2024

PONE-D-24-36446R1 

PLOS ONE

Dear Dr. Chtita, 

I'm pleased to inform you that your manuscript has been deemed suitable for publication in PLOS ONE. Congratulations! Your manuscript is now being handed over to our production team.

Kind regards, 

on behalf of

Dr. Prashant Singh 

Academic Editor

PLOS ONE